# Low-Dose Non-Targeted Effects and Mitochondrial Control

**DOI:** 10.3390/ijms241411460

**Published:** 2023-07-14

**Authors:** Dietrich Averbeck

**Affiliations:** Laboratory of Cellular and Molecular Radiobiology, PRISME, UMR CNRS 5822/IN2P3, IP2I, Lyon-Sud Medical School, University Lyon 1, 69921 Oullins, France; dietrich.averbeck@univ-Lyon1.fr or dietrich.averbeck@wanadoo.fr

**Keywords:** ionizing radiation, mitochondria, ROS, apoptosis, signaling, DNA damage response (DDR), low-dose effects, hormesis, adaptive response, hyper-radiosensitivity (HRS), genomic instability, cancer, bystander effects, non-targeted effects (NTEs), innate and adaptive immune responses, radiotherapy

## Abstract

Non-targeted effects (NTE) have been generally regarded as a low-dose ionizing radiation (IR) phenomenon. Recently, regarding long distant abscopal effects have also been observed at high doses of IR) relevant to antitumor radiation therapy. IR is inducing NTE involving intracellular and extracellular signaling, which may lead to short-ranging bystander effects and distant long-ranging extracellular signaling abscopal effects. Internal and “spontaneous” cellular stress is mostly due to metabolic oxidative stress involving mitochondrial energy production (ATP) through oxidative phosphorylation and/or anaerobic pathways accompanied by the leakage of O_2_^−^ and other radicals from mitochondria during normal or increased cellular energy requirements or to mitochondrial dysfunction. Among external stressors, ionizing radiation (IR) has been shown to very rapidly perturb mitochondrial functions, leading to increased energy supply demands and to ROS/NOS production. Depending on the dose, this affects all types of cell constituents, including DNA, RNA, amino acids, proteins, and membranes, perturbing normal inner cell organization and function, and forcing cells to reorganize the intracellular metabolism and the network of organelles. The reorganization implies intracellular cytoplasmic-nuclear shuttling of important proteins, activation of autophagy, and mitophagy, as well as induction of cell cycle arrest, DNA repair, apoptosis, and senescence. It also includes reprogramming of mitochondrial metabolism as well as genetic and epigenetic control of the expression of genes and proteins in order to ensure cell and tissue survival. At low doses of IR, directly irradiated cells may already exert non-targeted effects (NTE) involving the release of molecular mediators, such as radicals, cytokines, DNA fragments, small RNAs, and proteins (sometimes in the form of extracellular vehicles or exosomes), which can induce damage of unirradiated neighboring bystander or distant (abscopal) cells as well as immune responses. Such non-targeted effects (NTE) are contributing to low-dose phenomena, such as hormesis, adaptive responses, low-dose hypersensitivity, and genomic instability, and they are also promoting suppression and/or activation of immune cells. All of these are parts of the main defense systems of cells and tissues, including IR-induced innate and adaptive immune responses. The present review is focused on the prominent role of mitochondria in these processes, which are determinants of cell survival and anti-tumor RT.

## 1. Introduction

Non-targeted effects (NTE), such as radioadaptive effects, low-dose hypersensitivity, bystander effects (BE), and genomic instability effects induced by ionizing radiation (IR), are generally considered as being low-dose and low-dose rate effects [1,2,3,4,5,6]. A common feature of low-dose effects (<100 mGy) is the absence of strict dose proportionality [7]. Low doses are generally defined as up to 100 mGy, and low-dose rates are defined as up to 6 mGy/h [8]. However, bystander effects involving distant tissues or organs (so-called abscopal, i.e., out-of-field effects) may also be observed following high therapeutic doses (>2 Gy), as used in anti-cancer radiation therapy (RT) [9,10]. Bystander signaling is brought about by long-lived radicals, cytokines, interleukines, and other cellular mediators through intercellular gap junctions or freely by diffusion [11,12,13].

It appears that NTE involve cellular signaling and particular mitochondria-mediated signaling [14,15,16]. Moreover, mitochondria are at the origin of innate and adaptive immune defenses [17,18]. It is therefore not surprising that NTE are also related to mitochondrial functions affecting IR immune responses [19]. Interestingly, the biological consequences of NTE observed in cancer cells depend on the radiation quality: for example, Carbon ion radiotherapy (CIRT) clearly appears to be more effective than conventional RT [19].

Several mechanistically interesting phenomena are involved in low dose, NTE, and bystander effects of cells and tissues after IR involving mitochondria. There is signaling from irradiated and/or dying cells to unirradiated cells and communication of irradiated with unirradiated cells via the release of molecular mediators, extracellular vehicles (EVs), or exosomes containing mtDNA, nDNA, miRNAs, specific proteins, etc., and those recently put forward through the IR-induced UVA biophoton emission and modulation of mitochondrial oxidative phosphorylation (OXPHOS) [20], or, also, direct transfer of healthy mitochondria towards irradiated cells via tunneling nanotubes (TNT) [21,22] or autophagy or mitophagy [23,24,25,26].

Indeed, TNT belongs to a new type of bystander effect. It may involve the transfer of mitochondria through nanotubes between immunologically active macrophages and cancer cells. For example, it has been shown that intercellular nanotubes can mediate such mitochondrial trafficking between breast cancer cells and immune cells [22]. 

Furthermore, recent research has revealed that exosomes can act as important mediators not only for the induction of damage to neighboring unirradiated cells but also to distant cancer (metastatic) cells, and this can enhance conventional anti-cancer radiation therapy (RT) [27,28,29,30,31]. Such bystander effects can be mediated by mitochondrial DNA through exosomes like vehicles, too [10,32]. Exosomes affected by RT can be immunostimulatory producing systemic response and abscopal responses [10]. The cargo of such exosomes often contains mitDNA as well as nDNA and non-coding mi RNAs [33]. Mitochondria play an important role in NTE of IR. As demonstrated by Miranda et al., when comparing cytoplasmic-hybrid (cytobrids) cellular models derived from a human osteosarcoma cell line (148B) with wild-type mitochondria, with mutated mitochondria, and without mitochondria, no bystander effect was observed in cells without mitochondria, suggesting their involvement in IR-induced NTE [34]. Gorman et al. observed a significant transient increase in mitochondrial mutations (point mutations and deletions) in bystander cells associated with a significant decrease in mitochondrial membrane potential 24 h after exposure to conditioned media from irradiated tumor explants [35]. The effect disappeared 72 h after irradiation. Mitochondrial metabolism was altered in human rectal cancer in ex vivo explants through IR-induced bystander effects [36]. The rectal cancer secretome induced significantly higher ROS levels in bystander SW837 cells than bystander cells exposed to the secretome from normal cells. Recently, the role of biophotonic effects in NTE was put forward by Mothersill et al. [20,37,38,39,40]. Biophoton signals emitted from β-irradiated HCT116 p53 wt cells affected the activity of the mitochondrial Complex I (NADH dehydrogenase or NADH: ubiquinone oxidoreductase) and the impairment of ATP synthase function [20]. Clonogenic cell death could be observed after IR-induced biophoton signaling in bystander cells depending on their p53 status [37] and depolarization of the mitochondrial membrane [38], clearly indicating the involvement of mitochondria. Moreover, after a low dose (22mGy) of γ-irradiation, biophotonic emission could be observed from irradiated HCT116 p53wt cells [39]. In this context, the energy deposition by IR in cells is thought to lead to excitation decay processes with emission of biophotons that affect mitochondrial functions, such as energy (ATP) production, and induces the release of exosomes, thereby initiating bystander responses in other cells, i.e., adaptive responses, genomic instability, and cell death [40]. These effects may also include abscopal and cell-mediated immune responses [10].

Figure 1 schematically indicates the important involvement of mitochondrial reactions in metabolic regulation, intercellular communication, and immune reactions after IR.

The present paper focuses on low-dose radiation effects and non-targeted effects (NTE) involving mitochondrial functions. Adaptive radiation responses [41] and low-dose radiation hypersensitivity [42,43] are included because they are tightly linked to bystander effects as well as to both innate and adaptive immunological effects. Additional evidence is provided that non-targeted abscopal and immune effects observed at higher IR doses and during anticancer radiation therapy (RT) involve mitochondria. 

## 2. Low-Dose Radiation Effects

Since the atomic bombing of Hiroshima and Nagasaki in 1945, many epidemiological studies have shown that low-dose IR can have harmful effects, including various types of cancer [44,45,46], such as, to take a few examples, leukemia [47,48,49], lung cancers [50], leukemia and brain tumors in children [51,52,53,54], and, in nuclear workers (1945–2005), leukemia and solid tumors [55,56,57]. Radiation health risks were usually estimated using the Linear non-threshold (LNT) hypothesis model [58,59,60]. However, because of high variability and great uncertainties in the low-dose range, this seemed to not quite be satisfactory scientifically [61,62,63,64]. Large scale research programs (see, for example, the USA DOE program 1999 [65], http://lowdose.energy.gov, 6 December 2012 accessed 30 June 2023) as well as the more focused European program coordinated by MELODI (1999) on low doses < 100 mGy were launched in order to obtain a better scientific understanding of low-dose radiation responses [62]. Important classical and recent low-dose radiation phenomena were brought to light and investigated in depth, notably hormesis [66,67,68,69], radioadaptation [70], hyper-radiation sensitivity (HRS) [70,71], bystander effects and non-targeted effects [1,5,70,72], and genomic instability [5,70]. Importantly, these effects were found to be non-linear in the low-dose range of IR. In fact, mechanistic molecular studies revealed that important cellular processes were non-linear at low doses, such as the induction of genes (transcriptome) [73,74], gene expression and epigenetic controls [75], expression of proteins (proteome and phosphoproteome) [76], DNA damage signaling [77], intracellular signaling of cell cycle arrest, DNA repair and apoptosis [4], and connected intercellular and extracellular signaling [70,78,79]. 

An interesting aspect of intra-cellular signaling is the cytoplasmic-nuclear-shuttling of important proteins, such as ATM [80] and many others (NF-κB, Nrf2 etc. [81,82]. It can be noted here that ATM is an important phosphor protein, which is in the center of an impressive signaling network of integrative interacting proteins [83]. 

Recently, Maeda et al. have observed that irradiation (at <2 Gy) of the nuclei of V79 and WI-38 cells did not induce γH2AX but, rather, p73-dependent cell death [84]. However, when whole cells were irradiated, involving mitochondria and nuclei, γH2AX (i.e., DSBs) were induced and p21 upregulated, indicating cell cycle arrest. An inhibitor of ATM could suppress γH2AX foci formation. The authors concluded that at low irradiation doses, cytoplasmic IR modifies the ATM-mediated DDR and determines cell fate.

An important aspect of ATM activity is the ATM cytoplasmic-nuclear shuttling process that allows radiosensitivity (seen in IR tissue reactions) to be distinguished, radiation-susceptibility for the induction of cancer, and IR-induced degeneration and senescence [80,85]. According to these authors, the process proceeds as follows: IR exposure monomerizes ATM dimers in the cytoplasm, and the resulting monomers of ATM migrate into the nucleus, where they activate H2AX histones at DSB sites by phosphorylation, giving rise to γH2AX foci that can be visualized by antibodies and immunofluorescence analysis. In this way, IR-induced DSBs are detected. Most of them are repaired by non-homologous end-joining (NHEJ). During repair, two ATM monomers can re-associate to DSB sites to form autophosphorylated p-ATM, which is also detectable by specific antibodies and immunofluorescence analysis. Retardation in radiation-induced ATM nucleo-shuttling (RIANS) involving the delay of recognition of DSBs and their repair indicate a default in coping with DSBs related to either radiation sensitivity (tissue sensitivity) or radiation susceptibility (cancer). ATM is also involved in oxidative stress (redox) control, mitochondria, and autophagy in cancer [86].

From this, it becomes clear that a better understanding of these different processes, in particular, and the differences in channeling of signaling messages inside and among outside cells should provide valuable insights into the biological outcomes of low-dose radiation and the possible benefits or health risks.

Figure 2 lists the different facets of low-dose IR effects involving mitochondria. The role of mitochondria in these low-dose responses are further detailed in the following chapters.

Hormesis involves the induction of antioxidants and repair enzymes. Radioadaptation to high doses involves a priming low dose, inducing antioxidants and activation of some DNA repair. Low-dose Hyper-radiosensitivity (HRS) (at low doses <30 mGy) implies initial radiosensitization due to the absence of proper activation and signaling of cellular defense systems, such as antioxidants and DNA repair followed by IR resistance (IRR). Bystander effects and non-targeted effects (NTEs) refer to signaling between cells through mitochondria-derived messengers (radicals, ROS/RNOS, mtDNA, ATP, microRNAs, interferon, and cytokine released) via intercellular gap junction but also via Exosomes that can directly signal the damage via DAMPs to neighboring cells as well as to distant cells (abscopal effects). This may cause genomic instability, mutations, cell transformation, and cancer. ROS/RNOS and mtDNA, microRNAs, and ATP are stimulating the immune system. 

## 3. Hormesis

Hormesis is an example of beneficial low-dose IR effects [87,88,89] involving the boosting of mitochondrial functions [90]. Low doses of IR (<100 mGy) can indeed stimulate cell proliferation and cell survival, and they can upregulate antioxidant and immune defenses. This is a mechanism that reinforces cellular defense systems, which enable cells to cope with subsequent insults. In mice, it has been clearly shown that low IR doses (<100 mGy) are protective [91], whereas higher doses (>100 mGy) can be detrimental [92]. The role of IR-induced oxidative stress and ROS in this should be underlined [93]. Master switches such as Nrf2 and NF-κB are turned on. In particular, oxidative stress leads to translocation of Nrf2 from the cytoplasm to the nucleus and to the upregulation of antioxidant genes, such as GPx, Trx, CAT and Mn-SD [94]. Such protective mechanisms are apparently not activated in conditions of background radiation suppression, i.e., in deep underground laboratories [95,96,97].

Interestingly, senescence-related phenomena may also be inhibited by low dose IR, as recently shown by [98]. Using a drosophila Alzheimer’s disease model, amyloid-β1-42 induced cell death was alleviated by low-dose IR (50 mGy) via regulation of Akt and p38 pathways. Similarly, a low dose of 100 mGy caused the downregulation of neural pathways associated with cognitive dysfunctions in normal human aging and Alzheimer’s disease [99]. Recently, there were reports on individual cases of Alzheimer’s that showed significant recovery of their cognitive and their intellectual capacities after low (40–80 mGy) or even higher doses of IR [100,101,102]. It is thought that these beneficial effects of low-dose IR are due to an adaptive response mobilizing cellular defense and signaling systems (antioxidants, etc.). It will be of great interest to further elucidate the underlying mechanisms and to develop new efficient treatment modalities against Alzheimer’s on new solid scientific grounds. 

Moreover, it should be noted that mitochondria are involved in innate and adapted immune responses [17]. In fact, low-dose IR can boost mitochondria-dependent immune reactions together with activation of Nrf2 [103]. A dose of 200 mGy in Wistar rats caused a significant increase in antioxidant activity (catalase and glutathione-S-transferase) together with an increase in blood lymphocytes and eosinophils, suggesting activation of an immune response mechanism [104]. In addition, tumor growth after inoculation of S180 sarcoma cells in mice could be inhibited by 75 mGy of X-rays whole-body IR. Erythrocyte immune functions were improved, too. Furthermore, as reported by Lau et al. [69], thiocyanate, an antioxidant related to the immune system, was increased in adults’ saliva treated with low doses by cone beam computed tomography [105]. A low dose of 100 mGy γ-irradiation reduced BRAF V600E virus transformation of human thyroid normal follicular cells, and it also suppressed thyroid transformation in mice by restoring the thyroid metabolizing gene expression of PAX and, in addition, suppressed thyroid cancer carcinogenesis through inhibition of STAT3-miRNA-330-5p pathways [106]. As also seen by the previously reported low-dose IR effects on neoplastic transformation [107], the reduction of spontaneous lung cancers in mice [108], and immunogenic effects [109], typical low-dose hormetic effects exhibit beneficial outcomes. Interestingly, such hormetic responses could not be observed in malignant cells [110,111,112]. In line with this, after low-dose IR, ATM could initiate hormesis and the adaptive response in normal lung epithelial cells but not in lung cancer cells A549 [112]. In normal cells, the accumulation of Nrf2 in the nucleus via activation of ATM/AKT/GSK-3b signaling resulted in increased expression of antioxidants, which limited ROS production by high-dose IR [112]. Generally, adaptive low-dose radiation responses were absent in cancer cell lines in vitro and in vivo [113]. 

It should be noted here that cancer metastasis formation, mostly abscopal type effects, can also be inhibited by low dose IR [114,115,116].

As stated by Scott B.R. and S. Tharmalingham [117], the hormetic responses are not compatible with the linear-non threshold (LNT) model because of the non-linearity at low-dose IR, and LNT is also not considered to be compatible with low-dose non-targeted effects in carcinogenesis [118].

## 4. Adaptive Radiation Response and the Involvement of Mitochondrial Functions

Protective adaptive radiation processes are evolutionarily conserved [119]. Since the observations made by Olivieri et al. [120], the phenomenon of the adaptive radiation response is best described by the fact that exposure of human lymphocytes to a first low “priming” radiation dose using low concentrations of tritiated thymidine followed by a second higher challenging dose, i.e., 1.5 Gy of X-rays, induce less chromosomal aberrations than the high challenging radiation dose alone, and confers protection [121,122]).

Interestingly, pre-exposure to low-dose IR as also mild heat can trigger radioadaptation [123] and the heat-shock proteins, such as Hsp70, are involved in vitro and in vivo [124,125]. Pre-exposure with low LET IR followed by a high challenging dose of heavy ions also induced an adaptive response in human blastoid cells [126]. The authors concluded the possible upregulation of DSB repair. Nenoi et al. demonstrated radioadaptive responses affecting radiation-induced carcinogenicity in vivo (mice) [127]. 

Moreover, radioadaptive responses with a decrease in micronuclei (MN) and neoplastic transformation were also observed in C3H 10T1/2 mouse embryo cells [128]. They occurred in normal cells, too, although not in tumor cells [113,129]. Curiously, the radiation-induced adaptive response has been observed in cell lines showing HRS response (and the induced radioresistance, IRR) but not in cell lines showing bystander effects [130]. 

Specific radiation quality effects for adaptive responses appear to exist: teratogenic effects of heavy ion exposures (C, Si and Ne -ions ranging from 15–55 keV/μm) differed from those induced by Fe-ions (100 keV/μm) when preceded by a low X-ray dose [131,132]. With low doses of fast neutrons (high LET), a radiation adaptive response could also be shown in human fibroblasts [133].

There are epidemiological studies on human populations in low and high background radiation areas, as well, which indicate adaptive responses in low-dose high background areas in Yangjiang China, with lower cancer mortality here likely due to increased DNA repair and antioxidant capacity [134]. Individuals from Kerala exposed to > 5 mGy per year appear to be low-dose IR primed because they show less chromosomal damage in blood samples when challenged with 1 and 2 Gy than individuals from low normal natural radiation background areas [135].

In line with this, IR on the earth surface appears to maintain a certain level of radioadaptation, and the low natural environmental IR modulates high-dose responses. The response is clearly different in deep underground laboratories, where cosmic galactic, solar, and environmental radiation is absent (see the Gran Sasso National Laboratory in Italy) [25]. Such a low-dose environment for 4 weeks yielded a state of overgrowth with activation of p53, induction of apoptosis, and autophagic signaling. Radioadaptive responses have also been observed in blood cells from residents of high-level natural radiation areas of Kerala (India) [136].

Mechanistically, it is an important fact that low doses of IR (10–100 mGy) of γ-irradiation can stimulate overexpression of antioxidants, such as MnSOD, catalase and glutathione peroxidase [137]. Such activation of antioxidant enzymes was seen with priming doses 100 mGy and 20 mGy followed by a high challenging dose of 2 G and 3 Gy exposure, respectively [138]. Paraswani et al. showed that the adaptive response involved an increase in Mn-SOD catalase, thioredoxin reductase, glutathione peroxidase MMP, and metabolism together with increased translocation of the transcription factors NF-κB and Nrf2 to the nucleus [139]. This indicated that reactive oxygen species (ROS) are involved, and, also, nitric oxide radicals play a role [140]. This view fits the findings of Lall et al., who showed that in human fibroblasts, a low dose of IR induces a change of oxidative phosphorylation to aerobic glycolysis, conferring increased radio resistance, as can be observed in mice, too [141]. This is accompanied by upregulation of genes, such as glucose transporters, glycolysis enzymes. and the oxidative pentose phosphate pathway, and it involves downregulation of mitochondrial genes and, consequently, metabolic changes with increased glucose flux. Furthermore, the transcription factor HIF1α (hypoxia-inducible factor 1) is induced by low-dose IR. However, its transcription is stimulated by NF-κB, and its mRNA translation by the PI3/AKT/mTOR pathway. Using 100 mGy followed by an exposure to a high-dose 4 Gy of X-rays, the adaptive response depends on physiological oxygen concentrations (5%) [141]. When the radiation dose elicits a certain level of damage, p53 is activated, and the activity of HIF1α and glycolysis is diminished.

When applying, first, a small dose (0.1–0.5 Gy to primary human fibroblasts followed by a high dose, an adaptive response was observed with the induction of less γH2AX and different kinetics of DSB repair than with the high dose alone [142]. The adaptive response was not regulated in these primary fibroblasts by IL-6 and TGF-β.

However, the initial low-dose exposure led 24 h afterwards to the expression of cytokines GM-CSF (1.33×), IL6 (4.24×), IL8 (1.33×), TGF-β (1.46×) in the medium. The biological consequences of this are not yet clear, but these cytokines do play a role in cellular senescence [142]. An excellent recent review retraces the different facets of TGFβ radiobiology [143].

A threshold for the adaptive response was seen in pKZ1 mice [144]. The adaptive response could even be observed in ATM knockout heterozygous mice [145]. Curiously, very low priming doses (0.001, 0.01, 1 or 10 mGy) protected completely against chromosomal inversions induced by a single high dose of 1 Gy, and, also, against a proportion of spontaneously induced inversions. Thus, the response observed in the prostate and the spleen of mice was clearly nonlinear [146]. 

Generally, the dose range inducing the adaptive response is limited and thresholded. This does not support the LNT hypothesis for estimating low-dose health risks [64,88]. 

The cellular and molecular mechanisms of the radioadaptive response have been investigated by many authors [41,70,141,147,148]. The protective response of cells and tissues involves the transcription of specific genes, the activation of signaling pathways (i.e., the DNA Damage Response (DDR)), and cellular defenses stimulated by oxidative stress. It increases the detoxification of radicals, the production of antioxidants, DNA repair, protein synthesis, and the metabolic pathways supporting survival, stress responses, the endoplasmic stress response, protein folding, cytoprotection through autophagy, regulation of the cell cycle, reinforcement of immune responses (inflammatory/immunogenic), and suppression of genomic instability in animals and humans.

The main features of radioadaptive responses are schematically presented according to Guéguen et al. [148] in Figure 3.

Recently, it has been shown that radioadaptive response likely involves ATM cytoplasmic-nucleoshuttling [149]. Indeed, it seems possible that after a low dose of IR, just a few monomers of ATM will be formed (after oxidative attack of the ATM dimers in the cytoplasm) that are able to diffuse directly into the nucleus. These ATM monomers should be then able to easily recognize DSBs induced by the challenging high dose and promote their rapid repair. In fact, this could be observed in human cells of moderate radiosensitivity [149].

Thus, the mechanism of radioadaptive responses appears to involve the DNA Damage Response (DDR) and DNA repair and signaling (p53, ATM, PARP), the antioxidant Nrf2 pathway, cell survival/apoptotic pathways, endoplasmic stress (UPR), immune/inflammatory responses (NF-κB pathway), autophagy (mitophagy), cell cycle regulation (cyclin B1/CDK1 complex) [148], and the translational machinery [150]. The more selective type of autophagy, i.e., mitophagy, eliminating dysfunctional mitochondria, fulfils a protective function against mitochondrial oxidative stress (ROS) and apoptotic signaling [151]. In addition, mitochondrial nitric oxide (NO) can be induced by IR via Ca^2+^ -sensitive mitochondrial nitric oxide (NO) synthase [152]. NO can be regarded as an important signaling molecule. In particular, NO can initiate radioadaptive as well as bystander responses, thus providing evidence that there is a link between both phenomena [41].

## 5. Low Dose Hyper-Radiation Sensitivity (HRS) and Induction of Radioresistance (IRR)

Low-dose hyper-radiosensitivity (HRS) is a typical low-dose IR phenomenon (see, for review, [19,69]), and it is evidenced by many datasets [71]. It was first reported by Joiner et al. [153], and it is characterized by an early dip in the survival curves of mammalian cells, indicating hypersensitivity to IR at doses between 100–300 mGy followed by radioresistance (IRR) at higher doses [153,154,155]. HRS is not observed in all types of cells, and it can be quite variable in human and mammalian cells. In general, human cell lines that are quite resistant to 2 Gy do show HRS (for example, cell lines such as T98G, Be11, HGL21, and RT112). However, U373 glioma and SiHa cervix cells of similar high-dose radioresistance did not show HRS [155]. HRS has been observed using low and high LET IR in mammalian cells, but not at very high LET IR [156]. Liang et al. provided evidence for HRS occurring in human embryonic lung fibroblasts and lung cancer cells using doses between 20–100 mGy of X-rays [111]. Using the scheme of low < 0.3 Gy followed by higher doses 1 Gy and >1 Gy, HRS was clearly demonstrated (<0.3 Gy) as well as the induction of resistance [69,157]. This may well have an important bearing on the RT of cancer.

### 5.1. IRR 

IR-induced radioresistance (IRR) is usually observed after low doses of 100–300 mGy with increasing doses up to 1 Gy [69]. Such (IRR) radiation-induced radioresistance represents an obstacle for anticancer RT since the resistant cells are often proliferating rapidly and are quite aggressive [158]. In stem cells of colorectal cancer, the signaling pathways JAK2/STAT3/CCND2 are responsible for IRR. Inhibition of IRR can be observed when the non-coding miRNA TINCR is inactivated [158]. Other authors have found that cancer stem cells are more radioresistant because of their increased DNA repair capacity (use of altered repair pathways during S phase replicating DNA) [159] and other factors and signaling mediators, such as AKT, cyclin D1, A20/NFκB, ERK, JNK, ROS, and p53 [160].

Because of the likely importance of HRS in anti-cancer treatments by RT, a lot of research efforts were undertaken to find out the underlying mechanisms. As mentioned by Lau et al. [69], from early on, Marples et al. thought about a concept involving damage recognition, signal transduction, and DNA repair [157].

Firstly, the fact that HRS occurring at the low dose range 100–300 mGy followed by the induction of radioresistance (IRR) was interpreted as being linked to a low-dose adaptive phenomenon: the initial very low dose subsequently induced the repair of IR induced DNA strand breaks yielding radiation resistance (IRR) [153,161,162,163]. 

Secondly, Marples showed that low-dose HRS is likely to be associated to G2-phase cell radiosensitivity [162]. Indeed, in HRS, G2-phase cells enrichment and G2-checkpoint abrogation occurred [164,165,166]. The ineffective cell cycle arrest in IR-damaged G2-phase cells [162,166] probably led to the low-dose induction of apoptosis in HRS [167].

Thirdly, Marples et al. thought that HRS involves either recognition, signaling, or the repair of DNA damage at low doses, which should involve ATM, H2AX, 53BP1, and HDAC1 [157,162]. It could be expected that the initial low doses of IR induce some DSBs that cannot be repaired because they were not recognized by DNA repair enzymes causing the initial HRS. At very low doses, Rothkamm and Löbrich (2003) had noticed that the detection and DNA repair of DSB was somewhat compromised [77]. However, Wykes et al. could show that HRS was not due to a failure of DSB recognition. Instead, together with the failure of G2-checkpoint arrest, this suggested that a default in DNA damage response (DDR) signaling may be involved [168]. Xue et al. demonstrated that the regulation of DNA repair of DSBs by ATM determined the radiosensitivity of human cells to low-dose carbon ions exposures [169]. They demonstrated that ATR signaling cooperates with ATM in the mechanisms of low-dose HRS after carbon ion beam exposures [170]. Subsequently, at the higher doses, full ATM activation occurred together with DNA repair of DSBs associated with IRR [171]. In fact, 100 mGy γ-irradiation did not lead to activation of ATM by phosphorylation (4 h after irradiation). However, at the dose of 250 mGy, ATM phosphorylation increased fourfold (with increased DSB induction, as indicated by γH2AX induction), suggesting increased repair. 

Concerning the association of HRS and bystander effects, it could be demonstrated that intercellular gap junctions were involved in non-targeted bystander effects [12] and in HRS [170]. Burdak-Rothkamm et al. have already pointed out a link between ATM/ATR DNA damage signaling and bystander intercellular signaling [172]. In line with this, IRR was not observed in ATM-deficient cells from ataxia telangiectasia patients. Late oxidative stress induced by IR in G2/M phase cells as well as bystander effects (in cooperation with ATR) involved ATM [173]. 

### 5.2. Involvement of Immune Functions

Recent papers point to the involvement of immunological processes. Small doses (<0.5 Gy) of X- or γ-rays, protons, and carbon ions, π-mesons) elicited HRS in 80% of several mammalian cell lines [174]. However, using chronic exposures at low-dose rates, HRS was inhibited. The attachment of TGFβ3 to Alk1 was involved in this process, indicating an impact on immunological responses. According to Mothersill et al., TGFβ3 and p53 are agents involved in the transduction of bystander signals, with mitochondrial metabolism as a key factor for the final outcomes [7]. 

### 5.3. Involvement of Mitochondria

Maeda et al. observed that microbeam irradiation of the nucleus induced HRS but less when the whole cell was irradiated [175]. NO acted as mediator from irradiated cells to non-irradiated cells [176]. They suggested that irradiation of the cytoplasm could affect mitochondrial functions, and, in particular, the mitochondrial production of ATP and antioxidant enzymes. By the way, cytoplasmic irradiation by microbeam α-rays is known to cause dysfunction of mitochondria [177], and such dysfunction can activate mitophagy to maintain energy-homeostasis in cells [178]. Chandna et al. stated the importance of the nutritional and the physiological energetic state dependent on mitochondria in HRS [179].

As demonstrated by Ghosh et al., low-dose γ-irradiation (100–300 mGy) caused an increase in HRS in G2/M phase in human tumour cells and caused the drastic (−50%) downregulation of cellular adhesion proteins, such as connexin 43, a transmembrane protein involved in the formation of gap junctional channels (in bystander cells) [180]. This was not seen in normal human fibroblasts. Knockdown of Connexin-43 in tumor cells to similar low levels rendered tumor cells hypersensitive to low-dose IR and caused growth inhibition involving mitochondria-dependent apoptotic functions, such as change in mitochondrial membrane potential (MMP), cytochrome-C release, and caspase-3 activation. This provides clear evidence for the implication of mitochondria in HRS. 

Moreover, low-dose IR induced mitochondrial translocation of connexin-43 and siRNA-mediated depletion of connexin-43 stimulated pro-apoptotic mitochondrial events suggesting a cytoprotective role of connexin-43 in tumor cells through mitochondrial functions. Connexin-43 plays an important role in cancer and cancer progression [181,182]. For example, the in vitro mesenchymal-epithelial transition (MET) in NIH353 fibroblasts initiated low-dose HRS coupled with an attenuated connexin-43 response [183].

## 6. Bystander Effects: Short Distance NTEs 

Non-targeted effects such as bystander effects are brought about by intercellular communication. A low-dose of α-rays can cause genetic damage, such as sister chromatid exchanges (SCEs) in cell nuclei of cells that have not been actually hit by a-particles [1]. Bystander effects include the leakage of signaling molecules such as ROS, nitrogen oxide (NO), TGFβ, TNFα, IL-1, and IL-8 from the mitochondria of irradiated cells [184]. These mediators induce SCEs, micronuclei (MN), mutations, clastogenic, and lethal effects in neighboring so-called bystander cells [185]. Bystander effects have been observed after IR at low and high LET. Buonanno et al. were able to show that when analyzing low and high LET dose effects on co-cultured bystander cells cocultured with high LET, irradiated (iron or silicon ions) (151 or 51 keV/μm) cells for 20 generations exhibited less survival and more chromosomal damage, and protein and lipid oxidation correlated with decreased antioxidant levels (MnSOD, CuZnSOD), catalase and glutathione peroxidase (GPx), inactivation of redox-sensitive aconitase, and an increase in 41 mtDNA encoded (translated proteins) than those co-cultured cells exposed to low LET protons (0.2 keV/µm) [186]. Thus, regarding long-term consequences, NTE (bystander effects) greatly depended on radiation quality and dose, and it involved persistent oxidative stress arising from perturbed oxidative metabolism [5,19,70]. 

Moreover, bystander effects may include exosomes or extracellular vesicles containing nuclear and mitochondrial DNA, mRNA, miRNA, circRNA, lncRNA, and the cytokines TGFβ and IL-10 [29]. Interestingly, after exposure to heavy ions (carbon ions), expression of some miRNAs (miR654-3p et mi-378-5p) could be correlated with the therapeutic efficiency of these heavy ions [187,188]. 

Regarding the mechanisms involved, NTE bystander effects rely on intercellular communication using the intercellular gap junctions and mediating signaling proteins and molecules. The latter are released into the extracellular space and can damage neighboring cells (bystanders), and this sometimes, even at long distances, induces abscopal effects (up to 1mm distant cells in tissue) [189]. Bystander effects may stimulate replication, proliferation, energy metabolism of mitochondria, DNA repair, and immune responses [5,70]. Mitochondria are, via the emission of ROS, TGFβ, etc., implicated in apoptotic effects of NTE and bystander effects [19]. For example, late apoptosis could be induced by 50 mGy of γ-irradiation, including activation of p53, Bax, Bcl-2, and caspases 2 and 6 [190]. Portess et al. were able to show that 50 mGy of γ-radiation induced NTE in pre-cancerous and transformed 208Fsrc3 cells in the form of apoptosis via irradiated normal rat fibroblasts (208F) in co-culture, involving signaling of ROS/reactive nitrogen species (RNS) and the release of cytokines from damaged mitochondria [191]. 

In a well-documented review on bystander effects and intercellular communication, an interesting model has been put forward [192], based on the idea that, already in radiation targeted cells through DDR, the redox status is regulated in such a way that signals are already generated for neighboring cells. Through intercellular gap junctions, secreted factors, such as ROS and cytokines, diffuse towards and are reaching non-targeted neighboring cells, and this may, alternatively, be happening via the circulation. The recipient cells are reacting to the signals by changing their mitochondrial status and their cellular redox potential, inducing, in turn, oxidative stress and DDR-signaling responses, such as cell cycle arrest and, more often, replicating cells apoptosis [192]. This is in line with first observations of Azzam et al. [78], who noted that intercellular gap junctions are involved in the transmission of detrimental signals to non-irradiated bystander cells [130]. In addition, cells have invented other means in order to communicate to more distant cells and organs in the organism, too, such as, for example, via tunneling nanotubes (TNTs) and, additionally, via exosomes and extracellular vehicles.

In the present review, we underline the important involvement of mitochondrial reactions in metabolic regulation, intercellular communication, and immune reactions after IR (see Figure 1). 

As shown by Hei et al., cells deficient in mitochondrial DNA exhibited reduced bystander signaling affecting NO and Ca^2+^ signaling, pointing to the importance of mitochondria for bystander effects [16]. Inhibition of the activation of extracellular signal-related kinase ERK suppressed the bystander response, indicating an important role of mitogen-activated protein kinase (MAPK), which signals in bystander effects. NF-κB dependent gene expression of IL8, IL6, cyclooxygenase-2, tumor necrosis factor (TNF), and IL 33 in directly irradiated cells produced cytokines and prostaglandin E2, which, in turn, activated signaling pathways and also induced NF-κB dependent gene expression in bystander cells [193]. Interestingly, a Golgi protein, GOLPH3, was also shown to mediate IR-induced bystander effects via ERK/EGRA/TNF-α signaling [194]. In normal human fibroblasts, WI-38, γ-ray, and carbon-ion irradiation (up to 0.5 Gy) induced bystander effects mediated by nitric oxide (NO). The killing of bystander cells depended on the radiation dose, but not on radiation quality [195]. Irradiation of the cytoplasm yielded 53BP1 foci, i.e., DNA damage in directly hit and bystander cells [196]. ROS and RNS inhibitors did not prevent cytoplasmic irradiation induced damage, but did inhibit signaling to bystander cells. Functional mitochondria were necessary to generate the bystander signals due to the fact that bystander signaling was absent in cells lacking mitochondrial DNA [196].

A new type of intercellular communication along tunneling nanotubes (TNTs) has been found in eukaryotic cells, including natural killer cells, dendritic cells, T cells, endothelial progenitor cells, and prostate and malignant cells, moving along the nanotube path [197,198]. A few years later, Lu et al. were able to show that intercellular transfer of mitochondria via tunneling nanotubes occurred through tunneling nanotubes, which caused increased invasiveness in bladder cancer cells [199]. Spontaneous unidirectional transfer of mitochondria from T24 bladder cancer cells to less invasive RT4 cells was observed both in vitro (in transwell assays) and in vivo (xenograft tumor growth) [199]. Thus, intercellular trafficking of mitochondria between highly invasive and less invasive urothelial cells appeared to facilitate bladder cancer cell development, progression, and reprogramming. Invasiveness of bladder cancer cells included tunneling nanotubes (200 nm in diameter), promoting spontaneous intercellular mitochondria trafficking with subsequent Akt activation and mTOR signaling [199]. 

As reported by Gong et al., intercellular tunneling nanotubes are tubular structures (with a diameter 50–1500 nm and lengths of tenths and hundreds of microns) that can transport proteins, RNAs, viruses and organelles from one cell to another [200]. They can modulate cell death by delivering injured cells and by increasing the lysis of distant cells by long distance interactions between natural killer cells and target cells [198]. 

Moreover, mitochondrial transfer via microtubules was observed by Jin et al., who showed that ATM controls DNA repair and mitochondria transfer to neighboring bystander cells [201]. Using a co-culture system, they provided evidence for the transfer of mitochondria from healthy ATMwt to ATM−/− deficient cells and vice versa. In proliferating cells, ATM is usually nuclear. Recently, too, in post mitotic neurons, ATM has been mostly cytoplasmic, and has been associated with both oxidative stress and neurodegeneration [202].

They mention that activated ATM induces glucose-6-phosphate dehydrogenase (G6PDH), the rate-limiting enzyme of the phosphate-pentose pathway (PPP)––a pathway-producing mitochondrial NADPH for antioxidant pathways and nucleotide synthesis. 

Loss of the mitochondrial ROS-sensing function of ATM caused cellular ROS accumulation and oxidative stress in ataxia telangiectasia (AT) [203].

In addition, bystander-like effects involving whole mitochondria of immune cells have also been observed in anticancer radiation therapy (RT) [204], and, in fact, breast cancer cells revitalized themselves by sucking intact functional mitochondria from immunocompetent NK T cells via nanotubes to reduce immunological defense system. Wang and Gerdes showed that mitochondria could be transferred by tunneling nanotubes to rescue PC12 cells [21]. As another example, mitochondrial transfer induced by adipose-derived mesenchymal stem cell transplantation improved cardiac function in rat models of ischemic cardiomyopathy [205]. Fan et al. reviewed the pathophysiological significance of such a mitochondrial ejection from cells [206]. Very importantly, following UV exposure, single pheochromocytoma cells PCP12 could be rescued when they were cocultured with untreated PC12 cells involving tunneling nanotubes (TNTs) [21]. Single cell analysis revealed that microtubule-containing cells were formed by stressed cells (i.e., cells that started the first steps of apoptosis with loss of cytochrome c without activation of caspase 3) and promoted the transfer of mitochondria via tunneling nanotubes (TNTs) from healthy PC12 cells to stressed PC12 cells when co-cultured. The maximum speed reached by this mode of mitochondria transfer was about 80 nm/s, and was thus slower than that reported for axonal transport of mitochondria transport and neurons (100–1400 nm/s) [207]. Cells untreated with UV containing defective mitochondria did not rescue UV-treated cells. This proved that the transfer of mitochondria by TNTs was responsible for the rescue of UV-irradiated cells [21]. This is of interest because the preferential transfer of mitochondria from endothelial cancer cells through tunneling nanotubes could also modulate chemoresistance [208], knowing that mitochondrial transfer mitochondrial transfer can rescue aerobic respiration [209]. In axonal transport regulation of mitochondria tunneling microtubes, the nano-positioning and the tubulin conformation are important [210]. Indeed, mitochondrial transport impacts on synaptic homeostasis and neurodegeneration [211]. A unique conduit for the intercellular transfer of cellar contents (including mitochondria) is provided by tunneling nanotubes in human malignant pleural mesothelioma [212]. Weng et al. showed that human mast cells (MCs) can rapidly form TNTs (within 5 min), transporting mitochondrial and secretory granule particles with themselves or with cocultured glioblastoma cells [213]. This constitutes an “alarming” signal in inflammatory diseases, and it is important for immunological responses. Mitochondrial transfer and gene transfer studies were also undertaken in retina degenerative diseases [214]. TNTs may transport lysosomes and mitochondria in hematopoietic stem cell-derived macrophages (of a mouse) [215]. Mitochondrial transfer can also exhibit beneficial effects in neurodegenerative diseases [216]. In the chemotherapy of breast cancer, mitochondrial infusion appears to be quite promising as an anti-tumor therapy, leading to reduced glycolysis, increased oxidative phosphorylation (OXPHOS), reduced proliferation, and increased apoptosis [217]. The importance of mitochondrial dynamics as a new therapeutic target has also been put forward by Weiner-Gorzel and Murphy [218], and the roles of mitochondrial fusion and fission in breast cancer progression have been recognized [219]. For instance, modulation of mitochondrial ERβ expression inhibited triple-negative breast cancer tumor progression via activation of mitochondrial functions [220]. Indeed, therapies driving mitochondrial fission appear to be beneficial for breast cancer patients by suppressing signaling and metastasis [221].

## 7. Long Distance NTEs: Abscopal Effects

In RT patients, rare out-of-field effects, i.e., so-called abscopal effects, could be observed when a tumor received focalized IR exposure, and, at the same time, tumor regression was seen in distant unirradiated tumors [9]. These non-targeted effects consist of long distant bystander effects [13]. Demaria et al. were the first to demonstrate, in mice, that these effects were mediated by T cell activation and the immune system [222].

As indicated by Buonanno et al., bystander effects involve direct cellular interconnections via gap junction channels involving connexin as well as longer distance connections (300 μm apart) through tunneling nanotubes TNTs [13]. In the latter, mitochondria can be transferred in order to fuel distant cells. Systemic inflammatory responses can also be triggered by mitochondrial damage [223]. Concerning abscopal effects in cancer, RT of a lung tumor, in whole organisms signals, are propagated from the irradiated tumor to the unirradiated tumor or the healthy tissue regions through bystander effects. However, in addition, the irradiated tumor gives rise to the induction of systemic changes via bystander signals to contiguous cells through gap junctions and, also, to distant cells through tunneling nanotubes transferring mitochondria, lysosomes ions, and molecules. The irradiated cells are known to secrete soluble factors cytokines and chemokines, K+ and Ca^2+^ ions, and, of course, ATP (provided by mitochondria). Moreover, they may release extracellular vesicles (ECVs), transporting a variety of cargo (RNAs, proteins, and mitochondria) that can be transferred by the bloodstream. Thus, abscopal effect in distant organs and metastatic sites may involve soluble factors cyto- and chemokines, Ca^2+^ ions, and ATP with or without the immune system. However, the signaling may also affect progenitor cells, leading to genomic instability in the progeny. Genomic instability was dependent on the connexins expressed in the irradiated cells [224]. Following exposure to low or high LET-IR, the irradiated cells emitted signals to bystander cells via coupling by gap junctions, leading to micronuclei (MN) induction. Distant progeny of isolated bystander cells also showed increased MN levels. According to the authors, gap junctions composed of connexin26 (Cx26) or connexin43 (Cx43) mediate toxic bystander effects within 5 h of co-culture, whereas gap junctions composed of connexin32 (Cx32) mediate protective effects. However, the long-term progeny of bystander cells expressing Cx26 or Cx43 did not display elevated DNA damage, whereas those coupled by Cx32 had enhanced DNA damage. Hence, the outcome depended on the type of connexin coupling irradiated donor cells to bystander cells [224].

Such connexins and ATP play an important role in long range bystander radiation damage to the non-targeted unirradiated cerebellum of mice, too [225]. This work was recently extended, including the brain and the heart, showing long range bystander effects on both the brain and the heart after partial body IR involving miRNA, proteomic changes, and exosomes [226,227,228]. In cancer RT, the treatment schedule (fractionation) 3× 8 Gy could promote the abscopal tumor inhibition without affecting the humoral ant-tumor response [229]. 

After proton microbeam irradiation co-cultures of A-549 lung cancer cells, unirradiated bystander A-549 cells showed DNA damage, involving intercellular communication through gap junctions [230]. In contrast, irradiated A-549 did not affect normal human fibroblasts WI38 bystander cells in co-culture. In contrast, irradiated normal WI38 cells provided protective effects on A-549 tumor cells (independent of gap junctional intercellular communication) [230], which suggests an inverse protective signaling and rescue effect.

NTE can also be propagated by mitochondria DNA and RNA in vesicles similar to exosomes [28,32]. Exosomes may contain nuclear and mitochondrial DNA, cytokines such as TGFβ and IL-10, and, also, HGMB1, mRNA and circRNA, lncRNA and prostaglandin EZ (PGE2), ESCRT (protein transport), and heat shock proteins. They can constitute beneficial anti-tumor bystander or abscopal effects [9,30,231]. Cargo, such as nucleic acids, DNA, and RNA, as well as proteins, lipids, and metabolites, have been considered as mediators of NTE, including bystander effects and genomic instability [232]. 

However, the functional effects of IR modified exosome cargo in recipient cells are not yet fully understood. IR derived extracellular vesicles and exosomes play a vital role in intercellular communications, and they may even induce radioresistance and NTEs. In this way, they also contribute to RT outcomes [33]. As shown in breast cancer cells, MCF-7 exosome-mediated bystander effects of therapeutic dose of 2 Gy of X-rays increase the invasive potential, including the epithelial mesenchymal transition and the glycosylation, possibly involving miRNA and altered protein content cargo in the exosomes [233]. Thus, the induction and modification of exosomes by IR and their bystander effects are of increasing interest for RT anticancer treatments [234]. 

IR-induced bystander effects are schematically visualized in Figure 4.

Exosomes and also extracellular vesicles released from irradiated normal and tumor cells are intriguing because they differ in dynamics of secretion, in cargo, and in the radiation-induced bystander effects that have been observed [235]. They also differ in different cell types. Moreover, exosomes released from tumors appear to be specific, and they may therefore be useful for tumor diagnosis [31,234,235,236]. Evidence has been accumulating showing that radiation-induced exosomes exert bystander effects associated with radioresistance [31]. In the treatment of brain tumors by localized cranial RT, the central nervous system exhibits targeted NTE as well as abscopal effects [237]. Targeted IR effects in the CNS are due to radiosensitization of tumor cells involving regulation by microRNAs. NTEs are brought about by intercellular gap junction communication and exosomes with miRNA cargo transduced by intracellular endocytosis. This miRNA may regulate bystander and abscopal effects involving the Akt pathway in CNS tumor cells [237]. The CNS-derived exosomes and the miRNAs are able to cross the blood-brain barrier, and they may constitute useful biomarkers for therapeutic tumor responses [237]. 

In addition, so called “rescue effects” of irradiated cells have been observed, including retro-signaling of non-irradiated cells (»RIRE»)––i.e., in co-cultures unirradiated bystander cells assist irradiated cells via intercellular signal feedback, resulting in less cytotoxicity (apoptosis) and also in less MN formation in irradiated NHLF cells [238,239]. For example, bystander NHLF cells rescued cancer cells: HeLa cells exposed to 200 and 400 mGy of α-rays could be revitalized in co-culture with non-irradiated lung fibroblasts (NHLF). In co-cultures, bystander cells are in dialogue with the irradiated cells, i.e., a reverse bystander effects appear to exist [240]. In fact, when non-irradiated NHDF cells received signals from irradiated melanoma cells, the non-irradiated NHDF cells triggered rescue signals that modified the redox status of the irradiated melanoma cells likely involving mitochondria.

As recalled by Hamada et al., ROS play an important role in bystander responsessuch as survival, MN formation γH2AX foci, p53 and p21 up-regulation, and ERK ½ and JNK activation, as well as an increase in binding activity of NF-κB, AP1, and ATF2 [241]. Intracellular ROS in bystander cells have been quite persistent due to NAD(P)H oxidase, which can be induced by secreted TGF-β1 [241]. When released from irradiated cells, it causes γH2AX formation (DSBs) and changes in cell-cycle gene expression in bystander cells. As shown by Lyng et al., [242], the apoptosis-induced in bystander cells was associated with a decrease in mitochondrial membrane potential and increased intracellular Ca^2+^ levels. Calcium fluxes are known to modulate bystander effects [243]. 

Targeted cytoplasmic irradiation involving mitochondrial damage also resulted in bystander responses [244]. For bystander effects, mitochondria-dependent NFκB/iNOS/NO and NF-κB/COX-2 prostaglandin signaling pathways were important [245]. There is thus no doubt that functional mitochondria play an important role in bystander effects.

## 8. Genomic Instability

The phenomenon of genomic instability is characteristic of cancer cells and tumor progression [246]. It has been shown by Morgan [2,247,248] that low-dose non-targeted effects in vitro and in vivo included bystander effects and genomic instability, and that low-dose IR-induced mitochondrial dysfunction together with persistently elevated levels of ROS perpetuated genomic instability together with clastogenic and transgenerational effects [3,249,250]. Kim et al. found that mitochondria from genomic unstable cells are abnormal (partially dysfunctional) and contribute to persistent oxidative stress in unstable cell clones [250]. Dysfunctional mitochondria with mutations in complex II producing excessive mitochondrial O_2_^−^ and oxidative stress were clearly related to genomic instability [23,173,251]. Szumiel has emphasized the pivotal role of mitochondria in IR-induced oxidative stress, epigenetic changes, and genomic instability [252]. Evidence for links between IR-induced mitochondrial dysfunction and genomic instability, including epigenetic mechanisms, have also recently been highlighted [6].

Genomic instability of human peripheral lymphocytes has been seen after doses of 100–500 mGy with the induction of chromosome aberrations, induction of micronuclei at 100–2000 mGy, and, also, at a high dose of 2000 mGy induction of acentric chromosomes [253]. AG1522 cells co-cultured with HeLa α-irradiated (500 mGy) HeLa cells (expressing a specific type of connexin 32 channel gap junction) showed increased induction of genomic instability (induction of MN) in distant progeny cells after 24 population doublings [224]. 

Mechanistically, mitochondrial damage is involved in genomic instability by generating ROS, which, in human fibroblasts, affect the cell cycle signaling of Akt/cyclin D1 by the inhibition of protein phosphatase PP2A [173]. Mitochondrial ROS mediates genomic instability in low-dose IR in human cells via nuclear retention of cyclin D1. Chronic fractionated exposures to doses 10 and 50 mGy accumulate modified cyclin D1 in normal human fibroblast nuclei mediated by mitochondrial ROS [254]. ATM controls the cyclin D1 levels at a low-dose of IR. Accumulation of aberrant cyclin D1 perturbs DNA replication and perturbs replication of DNA, and it may cause induction of both DSBs and senescence. As noted by Shimura, ATM sensors ROS, and it has a role in maintaining genomic integrity [255]. Deficiency in ATM causes nuclear genomic instability and oxidative stress. AT patients and ATM-deficient mice exhibit oxidative stress and mitochondrial abnormalities. Shiloh has also pointed out that ATM not only plays a role in genomic instability but also plays a role in the cerebellar degeneration of ataxia telangiectasia (AT) [256]. In fact, the role of ATM is intriguing: ATM is associated with DDR signaling, the control of cellular redox balance and mitochondrial function, immunodeficiency, chronic lung disease, cancer predisposition, endocrine abnormalities, segmental premature aging, and radiation sensitivity. In line with this, a recent paper by Mitiagin and Barzilai describes the role of ATM in cerebellar pathology, and it emphasizes the role of ATM in maintaining the cellular homeostatic redox state [257]. It is now recognized that ATM has a wide-ranging protective role involving nuclear damage but also cytoplasmic regulation [258]. Undoubtedly, ATM signal transduction affects mitochondrial radiation responses in terms of ROS production and control [255], and this is determining ROS-mediated genomic instability, the tumor microenvironment, and immune responses. Mechanistically, the function of ATM is related to mitochondrial maintenance and turnover, and, moreover, even to the regulation of protein homeostasis [259]. According to Valentin-Vega et al., the AT syndrome characterized by the loss of ATM functions might be considered as a mitochondrial disease [260,261]. 

As has been shown by Stagni et al., the activation of ATM in the cytosol by ROS and hypoxia plays an essential role in the regulation of autophagy [262]. ATM inhibits mTORC1 in hypoxic conditions and regulates HIF-1. In the presence of ROS, ATM regulates peroxisome degradation (pexophagy) via (phophorylation of Pex5) and mitophagy (modulation of Beclin-1) [262]. Autophagy restricts the mitochondrial DNA damage-induced release of endonuclease G to regulate genome instability [263]. There is evidence that oxidative activation of ATM can take place in mitochondria and not in peroxisomes [264]. In AT, a fraction of ATM proteins is localized in mitochondria and is rapidly activated by mitochondrial dysfunction [260], and ATM is involved in the regulation of mitophagy [261].

On the other hand, apart from its prominent role in the DDR, ATM kinase could finely tune the balance between senescence and apoptosis: activated ATM promoted autophagy (mitophagy) and sustained the lysosomal-mitochondrial axis, promoting senescence but inhibiting apoptosis [262].

An interesting observation has also been reported by Fakouri et al. concerning the interrelationship between genomic instability and mitochondrial dysfunction in mammalian cells and its importance for age-related functional decline [265]. Poly (ADP-ribose) polymerase I (PARP1), when persistently activated, depletes cellular energy reserves, resulting in mitochondrial dysfunction, loss of energy homeostasis, and altered cellular metabolism. Mitophagy of dysfunctional mitochondria could help to preserve human health. Any persistence of DNA damage perturbs mitophagy via the NAD+-SIRT1-AMPK pathway. Signaling pathways also activated by DDR interfere with mitophagy, including PARP1 activation and both NAD+ depletion and ATP depletion. This, in turn, leads to increased mitochondrial activity, with increased ROS production and a decrease in mitophagy associated with increased cell death [265]. In addition, long lasting epigenetic changes are driven by mitochondrial dysfunction [266].

How far autophagy and mitophagy affect IR induced immune responses is still an open question, but autophagy and mitophagy as well as ATP depletion should certainly modulate innate and adaptive immune responses.

## 9. Mitochondria and Innate and Adaptive Immunity Induced by IR

In recent years, it has become evident that mitochondria play an important role in the intra- and intercellular communication together with NTE and bystander responses, and they are essential for both innate and adaptive immune responses. This also holds true for general IR responses and the outcomes of IR exposures. In the following section, this is highlighted for low-dose IR effects and also for higher dose RT effects.

Mitochondria are considered to be the central hub of the immune system [267]. In eukaryotic and mammalian cells, they govern responses from outside aggressions and stress and control innate and adaptive immunity [17,268]. 

This capacity is part of a general defense mechanism, which is thought to stem from endosymbiotic bacteria—i.e., to originate from a last eukaryotic common ancestor (LECA) subjected to endosymbiotic pressure combining with an endosymbiotic alphaproteobacterial partner (for example, Asgard archaea 0.3%), giving rise to the first eukaryotic common ancestor (FECA) [269]. 

The immune system highly depends on mitochondria, which supply the energy requirement and maintain the system activation with the production of ROS and important metabolites [270]. Usually, the innate system responds first to alarms from injured cells, including the release of mtDNA from cytosolic escape. Thereafter, the adaptive immune system joins the subsequent inflammatory reaction, orchestrating different types of T cells and their activators (including antigen-presenting cells), co-stimulating molecules and cytokines. Extracellular mitochondria can help regenerate and activate immune cells to eliminate damaged cells. Nanotubal transfer of mitochondria has been observed in many instances [271]. It would be interesting, indeed, to elucidate the mechanisms of IR on nanotubal intercellular transfers (TNTs) of mitochondria between cancer and immune cells. For example, it could well be that high LET IR is more effective in destroying TNTs than low LET IR. For example, carbon ion RT (CIRT) could be expected to very efficiently block the sucking of mitochondria cells from immune competent cells by breast cancer cells (as observed by Saha et al. [22] due to the destructive power of the high-density ionization tracks of carbon ions [272].

### 9.1. Low-Dose Immune Effects

Low-dose effects on the immune system have been recently reviewed by Lumniczky et al. [273], and they support the idea that low-dose IR (<100 mGy) is associated with pro-inflammatory responses, that intermediate dose IR (100 mGy–1 Gy) is associated with anti-inflammatory non-linear responses on chronic inflammatory conditions, and that high-dose IR (RT) (>1 Gy) is associated with pro-inflammatory responses and immune suppressive effects. However, there are some recent reports that underline the capacity of IR to elicit beneficial immunogenic responses, such as, for example, immune defense reactions coping with infected or cancer cells. As will be seen below, the mitochondrial reactivity and functions play an important role in these cellular immune responses.

An example of low-dose IR induced immune reactions is illustrated in Figure 5, according to Cho et al. [274]. In mice, low-dose IR (50 mGy) induced transcription of genes that are involved in immunogenic responses involving the mitochondrial NADH dehydrogenase (subunit of complex I) and the subunits of ATP synthase, as well as energy metabolism plus cytokine gene expression in CD4+ T cells [274]. 

Upregulated gene sets in CD4+ cells included the mitochondrial envelope, the inner and outer mitochondrial membrane, the respiratory chain, and ribosomes. In general terms, genes associated with RNA translation, mitochondrial function, cell cycle regulation, and cytokine activity increased after 50 mGy low-dose IR in CD4+ cells that underwent activation. Cytokines produced by T helper 1/2 cells (activating macrophages and helping B-cells for the production of antibodies and the development of cytotoxic T-lymphocytes) (IFN-γ, IL-4, Il-5) were upregulated, whereas those from T regulator (Treg) cells (TGFβ1, TGFβ3) were downregulated [274]. While the first part corresponds to the activation of CD4+ cells and the innate immune response corresponds with pro-inflammatory cytokine production, the second part includes the activation of CD8+ T cells corresponding to the adaptive immune response, which determines the final beneficial immunogenic outcome. In line with this, low-dose IR (75 mGy) was shown to be directly able to activate natural killer (NK) cell proliferation in vitro [275]. 

Furthermore, it is of interest that low-dose RT (<1 Gy), such as the intermediate dose of IR (0.5 Gy), could induce favorable T cell conditions for beneficial antitumor effects, although at a higher dose, the effect did decline [276]. The intermediate dose of (0.5–0.7 Gy) used in RT was also shown to reduce inflammatory and degenerative diseases [277,278] involving the expression of the X-linked inhibitor of apoptosis (XIAP) and TGFβ1 as well as reduction B and L-selectin and, also, secretion of the cytokine IL-1β or chemokine CCL20.

A recent review paper by Dove et al. reported that low-dose RT could attenuate osteoarthritis via modulation of mitochondrial function and anti-inflammatory activity [279]. For instance, exposure to doses between 0.5–1.5 Gy caused a decrease in the expression of inflammatory factors, such as MMP13POSTN and ADAMTS5 [280].

After β-radiation (51Cr) exposure of mice, natural-killer group 2 member D (NKG2D) cells were upregulated in the presence of p53. This led to a potent activation of NK and stimulated CD8+ T cells to attack tumor cells [281]. RT also combined with IL-15 increased expansion of NK cells, and CD8+ T cells mediated antitumor immune responses [282,283].

In view of the anti-inflammatory and immunogenic potential of low-dose IR, during the recent COVID-19 epidemic, low-dose RT (30–150 cGy) has been suggested to cope with the cytokine storm associated with the severe pneumonia of COVID-19 infections [284]. Indeed, low-dose RT attenuated ACE2 depression and inflammatory cytokines induction by COVID-19 in human bronchial epithelial cells [285].

It is important to note that all of these reactions and immune responses are dependent on mitochondrial functions and sufficient energy (ATP) supply [267,286]. Immune cells depend on the energy supply provided by mitochondria adenosine triphosphate (ATP) in order to grow, differentiate, and perform [287,288].

### 9.2. Role of mtDNA in Innate and Adaptive Immune Responses

IR is well known to induce free radicals and free electrons, giving rise to the accumulation of reactive oxygen species (ROS) in cells and tissues [14]. Among the ROS induced by IR are hydroxyl radicals, resulting in oxidative base damage and single and double strand breaks (SSBs and DSBs) in DNA. It is very important to note that mtDNA oxidized by hydroxyl radicals exhibits increased immunogenicity [289]. 

As shown in the paper by Nadalutti et al., the small circular mtDNA, encoding for just a few genes (37 genes that are involved in oxidative phosphorylation), relies on nuclear DNA genes to maintain and repair its own DNA [268]. However, mtDNA is in the center of cellular energy (ATP) production), stress, and innate and adaptive immune responses. ATP is produced in mitochondria by oxidative phosphorylation via the electron transport chain (ETC) and the tricarboxylic acid cycle (TCA) pathway [290]. Usually, mtDNA is present in several copies in mitochondria so that it is the ratio between functional and dysfunctional copies as well as the number of mitochondria per cell (mitochondrial dynamics) that determine final outcomes at the cellular and tissue level. 

Already at low doses of IR mtDNA is damaged and gives rise to enhanced ROS production and leakage of O_2_^−^, even if not yet significant numbers of DSBs are induced in nDNA [291]. Mitochondria containing damaged (oxidized) mtDNA can be eliminated by mitophagy in order to re-establish normal cellular functions [290]. Prolonged persisting oxidative damage to mtDNA is accompanied by the decrease or loss of mitochondrial membrane potential, and also by mitochondrial outer membrane permeation (MOMP), which leads to apoptosis. In fact, mtDNA is more sensitive to oxidative damage than nuclear DNA, since it is not protected by histones, contains more CpG sites, and is also only slowly repaired by base excision repair [292]. Mitochondria and mtDNA contribute importantly to bystander responses, too [196,293].

Recent experiments with a mitochondria specific photosensitizer revealed that oxidative damage very quickly induced damage to mtDNA and mitochondrial dysfunction (loss of respiration, decreased electron transport chain (ETC) activity, mitochondrial fragmentation), but, at first, nuclear DNA (nDNA) damage was absent. Only several hours later (48 h), a persistent wave of further oxidative damage involving O_2_^−^ and H_2_O_2_ was observed, which also damaged nDNA. Subsequently, it was shown that during apoptosis, BAK/BAX macropores in the mitochondrial outer membrane (MOM) facilitated herniation as well as mtDNA efflux [294] and mtDNA- dependent immune reactions. 

Similarly, IR exposure increases calcium influx and mitochondrial ROS, resulting in the release of mtDNA into the cytosol [17]. This can activate the NLRP3 inflammosome involved in inflammatory responses. The mitochondria, and, in particular, the mitochondrial outer membrane (MOM) act as a platform for the innate immune response for pro-inflammatory responses as well as for the subsequent steps of adaptive immune and immunogenic responses. In the absence of apoptotic caspases, the proteins, BAK and BAX, trigger the release of mtDNA, which binds to cGAS and catalyses the production of cyclic GMP-AMP, i.e., cGAMP, which, in the cytosol, binds to STING at endoplasmic reticulum-mitochondria contact sites, promoting the induction of type I INF transcription and immune responses. 

Damage to mitochondrial DNA can produce autophagy, cell death, and disease [292]. This involves the accumulation of certain types of mitochondrial damage, triggering cell death in the absence of DNA ligase III (Lig3) or exonuclease G (EXOG), which are enzymes that are required for repair [292]. Interestingly, as reported by Nadalutti et al., human disorders, such as ataxia telangiectasia (AT), Alzheimer’s disease, and also neurogastrointestinal encephalomyopathy, are associated with decreased ligIII levels, loss of mtDNA integrity, and mitochondrial dysfunction [268].

### 9.3. Immunogenic Effects in Antitumor RT

In recent years, it has become clear that the efficacy of RT in anticancer treatments is not only due to its great effectiveness at inducing oxidative lesions, DNA single and double strand breaks, and complex lesions in nuclear DNA that are difficult to repair, but is also due to the capacity of IR to induce damage to mitochondria mediating innate and adaptive immune responses [17,19,268]. In other words, the curative effects of RT are brought about by direct cytotoxic effects on tumor cells and by the reprogramming of the tumor environment (TME), initiating an antitumor immune response and immunogenic cell death. The innate immune system is turned on by cellular damage affecting mitochondria, followed by the production and the release of cyto- and chemokines to neighboring cells (TME), which promotes the infiltration of dendritic cells (DCs), macrophages, cytotoxic and regulatory T cells (Tregs) [17,295]. Myeloid-derived suppressor cells can also be activated. In this way, RT is remodeling the tumor cell microenvironment in order to give rise to beneficial or adverse outcomes [296].

Intercellular communication is very important in the tumor microenvironment (TME). Suppressive immune reactions may be brought about by immunosuppressive MDSCs and Tregs and by immunosuppressive cytokines (IL-10, TGFβ). However, depending on tumor type, radiation dose and schedule RT can stimulate antitumor immune reactions. Recently, it has become evident that RT can produce necrotic and apoptotic cell debris, and can damage associated molecular patterns (DAMPs) from tumors that can serve as tumor-associated antigens. When captured by antigen-presenting dendritic cells, they can be recognized by specific CD8+ T cells, which then destroy primary and metastatic tumors [10]. 

Exosomes that play a role in bystander and NTEs can help in this immune cell trafficking because DCs-derived exosomes can mediate CD8+ T cell activation. The programmed death-ligand 1 (PD-L1) (CD274) is found upregulated in many cancers, and it binds to PD-1 (CD279) on T cells, which inhibits T cell activation. PD-L1 is present on immune cells (DCs, macrophages and myeloid-derived suppressor cells (MDSCs)) but is also present on (and in) exosomes that are involved in bystander and abscopal effects. Apparently, exosomal PD-L1 can exert its immunosuppressive effects on many types of cancer cells, and this is of great therapeutic value in anticancer RT [10].

In fact, local RT induces a prooxidant state and pro-inflammatory reactions that can trigger both innate and adaptive immune responses [10,297]. This oxidative stress leads to the release of metabolites, calreticulin, heat shock proteins (HSP70, HSP90), ATP, HGBM1, nDNA, mtDNA, RNAs, and lipids and cytokines that act as damage associated molecular patters (DAMPS), promoting innate anti-tumor immune responses [298,299,300]. 

After local IR, DAMPs from the damaged or stressed tumor cells bind to specific pattern recognition receptors (PRRs) expressed on DCs [300]. DAMPS are involved in immunogenic cell death [297]. PRRs on DCs may include Toll-like receptors (TLRs) (for example, the TLR-4 receptive for viral or IR attack), c-type lectin receptors recognizing extracellular stimuli, cytosolic PRPs (such as retinoic acid inducible gene gene 1 (RIG-1)), receptors (RLRs) sensing RNAs, DNA sensors (cGAS-STNG, AIM2), and NOD-like receptors (NLRs) for intracellular pathogens, or DAMPs (HMGB1, ATP, NAD+ and adenosine) from tissue injury. Their binding induces dendritic cell (DC) maturation and the presentation of antigens, and promotes adaptive immunity in tumors via T-cell priming [301]. RT also induces cytosolic nucleic acid-sensing (cGAS-STING-dependent) pathways triggering type I interferon (IFN) [301,302]. Indeed, following IR of tumor cells, the cGAS-STING-IRF3-type I IFN cascade activates DCs, which leads to activation of the adaptive immune response via priming of downstream effector cancer-specific T cell recognition (CD8+) as well as lysing tumor cells locally and at distant sites [299,300,303]. Thus, with RT of localized tumors, immunogenic cell death, robust tumor regression, and the destruction of distant tumors can be achieved [301]. 

A schematic view of the steps of the immunogenic response is presented in Figure 6.

The induction of DNA damage at high doses (>1 Gy) of IR stimulates the DNA Damage Response (DDR) and can alert localized or systemic host immunity, and vice versa, via DNA damage signaling. Indeed, the IR-induced DNA damage in cells can lead to the activation of a cytosolic DNA sensing mediated by cyclic GMP-AMP (cGAMP) synthase (cGAS) and the stimulator of interferon genes (STING) [304]. This triggers type I interferon (IFN I) signaling towards DCs (see Figure 6). In this way, the innate and adaptive immune responses are initiated one after the other. After RT, the DNA damage is signalized through cGAS-STING, leading to the activation of CD8+ cytotoxic T cell-mediated tumor destruction. Importantly, genome instability to innate immunity are linked via cGAS surv eillance of DNA damage in the form of micronuclei (MN) [305]. 

RT may also induce the release of chemokines that recruit effector T cells, which can attack inflammatory tumor tissue. Using a breast cancer mouse model, Matsumura et al. observed that IR increased the secretion of C-X-C motif chemokine ligand 16 (CXCL-16), which could bind to C-X-C motif receptor (CXCR6) on th1 cells, activating CD8^+^ T cells. As a result, the CD8(+) CXCR6(+) T cells were found in the breast tumor tissue [306].

As described by Lin et al., following RT, dying lung tumor cells can release tumor-associated antigenic factors that are recognized by antigen-presenting cells (APCs), such as DCs, and subsequently activate CD8+ T cells [307]. These cells drive specific immune responses, and they target the primary lung tumor as well as the metastatic tumor cells [307]. After IR exposure, activated tumor specific T cells (CD8+) exit from the lymph nodes and circulate through the body from the irradiated tumor area to as yet unirradiated areas, eliciting NTEs and distant bystander (abscopal) effects [298,308]. 

Thus, the immunogenicity of RT is quite high and promising. However, it has to be kept in mind that the antitumor immunity may also be suppressed by regulatory T cells (Treg cells) [18,287,288,309]. However, RT doses and schedules exist that allow for increased tumor infiltration by effector T cells, as they can deplete or inactivate immunosuppressive Treg cells [309].

As observed in mice, during RT, the dose per fraction rather than the biological effective dose appeared to determine the induction of CD8+ T cell activity, whereas the induction of natural killer (NK) cell activity required a high effective dose independent of the treatment schedule [18].

In fact, the dose per fraction was important for the accumulation of Treg cells within the irradiated tumors. For AT3-OVA tumors, the RT induced response of Treg cells played a decisive role in the activation of adaptive immunity in a dose per fraction-dependent manner and also in the early activity of NK cells after RT. After Treg cell depletion by an antibody (9H10), only the control of tumors outside the irradiated volume (abscopal effects) and memory cell responses could be observed. Interestingly, in MC38 tumors, Treg cell enrichment was absent after RT but effective CD8+ cell activation occurred [18].

The induction of immune responses by IR is a very complex matter [10]. Radiation quality and dose as well as fractionation regimes and dose rate may all determine the immunogenicity of tumors and the possibility of inducing antitumor immunity for a particular tumor. IR induces changes in the tumor microenvironment and in intercellular communication, i.e., near and distant NTE and bystander effects. Immuno-suppressive effects can be IR-induced by high doses when there is low tumor immunogenicity and low tumor antigen production associated with the induction of immunosuppressive cytokines, such as IL-6 and TGFβ, in the TME. A certain IR dose threshold appears to exist that can shift the balance towards the activation of immunogenicity. At low LET IR, low doses may favor the activation of innate immune responses and allow for immunogenic responses, whereas high-dose effects on the immune system may rely more specifically on radiation quality and on both the type of tumor and the TME set-up (the availability of healthy immune cells) in order to turn on the adaptive immune responses. The differences in immunogenic effects of RT using low LET (X-γ-rays, photons) or high LET IR (heavy ions, carbon ions, etc.) are probably due to their capacity to induce well-targeted and complete apoptosis in tumors cells without much possibility of reversal, leaving intact immune cells in the tumor microenvironment (TME) that are able to respond to the highly immunogenic signals from the dying cancer cells [19].

Thus, immune responses rely on mitochondria due to their important role in energy supply and metabolic regulation, their role in initiating and directing intra- and intercellular signaling, their role in the boosting of the cellular defense systems, and, finally, their role in mediating bystander effects and NTEs (including abscopal effects). 

IR-induced damage of cells induces mitochondrial damage and apoptosis, with DAMPS leading to the activation of dendritic cells. Activated dendritic cells sense mtDNA from apoptotic cells and mediate activation of macrophages and cytotoxic CD8+ T cells, which destroy the local tumor as well as the distant metastatic tumor cells (abscopal effects). The chain of events on tumors include the induction of nuclear and mitochondrial damage, bystander and nontargeted effects (NTEs), activation of effects on distant tumors, and metastases via the induction of inflammatory cytokines as well as the activation of immune cells diffusing or migrating to unirradiated distant tumor sites. This shows that IR and RT can induce immunogenic cell deaths of both local and distant tumors.

## 10. Concluding Remarks

Because of their microbial origins, mitochondria are very reactive organelles that sensitively sense external and internal insults of cells of physical (radiation), chemical (metals), or biological (viral, bacterial, and fungal) origin [269]. Insults of any kind need intra- and or intercellular signaling and communication to elicit a coordinated cellular reaction, ensuring homeostasis repair and rescue (with senescence) or removal by autophagy or self-destructive apoptosis [292]. Their extremely high reactivity is due to receptors on the cell membrane and mitochondrial membrane receptors that interact with each other. Part of these are involved in cellular defenses: antioxidant antiradical defenses, DDR controlling DNA repair, cell cycle arrest and apoptosis, and immune reactions [18].

The largest part is involved in mitochondrial energy supply and cellular metabolism. Electron leakage during energy metabolism leads to the generation ROS and NOS [15]. Mitochondrial ROS are usually 10 times higher in tumor than in normal cells [310]. To avoid excessive oxidative damage, cancer cells activate potent cellular antioxidant systems in order to counteract ROS by superoxide dismutases (SODs) enzymes in mitochondria, catalyzing O_2_^−^ to H_2_O_2_. These can be reduced to H_2_O by catalases (CATs), glutathione peroxidases (GPXs), and peroxiredoxins (Prxs) [310,311]. 

The highly diffusible reactive oxygen or nitrogen species (ROS and NOS) are formed during metabolic activity and are involved in redox-regulated signaling, and they are second messengers in cell signaling and direct gene expression. Indeed, oxidation reactions promote activation of protein kinases, whereas phosphatases and zinc finger proteins are inactivated. Transcription factor activation of NRF2 ((nuclear factor erythroid 2–related factor 2) is increased by reduction reactions. Redox sensitive reactions of signaling proteins are often reversible to allow switching and adjustment of metabolic activities. The levels of antioxidants determine the outcomes of IR [15]. 

Moreover, important structural differences exist between mitochondrial DNA (mtDNA) located in the inner membrane space of mitochondria and nuclear DNA. Mitochondrial DNA is more radiosensitive due to the lack of protective histones and limited DNA repair [292]. Furthermore, mtDNA can be released by dying cells, and it is part of damage-associated molecular patterns (DAMPS) [297], which are very immunogenic due to the presence of CpG isles.

Since mitochondria constitute the energy platform [252,267,286,290], the first reaction after IR exposure is a modification of the energy metabolism. Generally, metabolically very active cells, such as cancer cells, contain more mitochondria and more mtDNA than normal cells. Mitochondrial mass varies in different species and animal organs, and is mostly related to their proliferative capacity. IR exposure at low (5 mGy) and high doses (5 Gy) can produce increases in mitochondrial mass [312]. 

Low doses induce already small but significant changes in oxidative phosphorylation, resulting in leakage of ROS and NOS as well as changes in metabolic ATP production. This affects the energy available to the targeted cells and to other bystander cells. A special case is tumor cells being in interaction with the tumor micro-environment.

Klammer et al. show that there are many factors involved in radiation-induced bystander (RIBE) and NTE effects, and the involvement of mitochondria related functions and bystander and innate and adaptive responses is striking [192]. ROS and mitochondrial generated ROS are very important, and they determine the oxidative state and oxidative stress level induced. Modulation of oxidative stress by IR is crucial for the bystander and also for the immunological responses. Some signaling cascades (Calcium fluxes, MAPK, and NF-κB networks) participate in both phenomena, and appear to be associated. Bystander effects can perpetuate oxidative stress induced damage and also immunological (inflammatory) responses, which may continue to perpetuate damage. Oxidative stress and oxidative damage are highly damaging, and they drive immunological responses through DAMPS and PAMPS [223,313]. According to Kong et al., mitochondrial ROS and ATP can be considered as damage-associated molecular patterns (DAMPs) that may give rise to systemic inflammatory responses [223]. 

In normal cells, ROS are produced at low concentrations and are effectively neutralized by the potent antioxidant systems of the cells [314]. A moderate increase in ROS levels by chronic oxidative stress and LDIR induces random mutations in cells and promotes tumor cell proliferation, metastasis, and radioresistance. Moderate ROS may increase radioresistance of cells in RT by triggering adaptive hometic responses and promoting autophagy [315] or triggering apoptosis independently of DNA damage [314,316]. ROS basal levels are often higher in tumor cells than in normal cells [317]. Low to moderate ROS levels act as signal transducers, activating cell proliferation, migration, invasion, and angiogenesis, whereas high ROS levels damage proteins, nucleic acids, lipids, membranes, and organelles associated with cell death [317]. Manipulations of IR-induced mitochondrial ROS are promising in RT and immunotherapies. 

### 10.1. RIBE and NTE and the Role of Mitochondria

A number of valuable reviews describe and comment on the importance of ionizing radiation-induced bystander (RIBE) and nontargeted effects (NTE) in ionizing radiation responses concerning, initially, the effects of low-dose ionizing radiation (LDIR) (<100 mGy) and the effects of both moderate- and high-dose (RT) exposures in recent years. At first, the bionegative effects of LDIR-induced bystander effects, such as cytotoxicity [78,312,318,319], mutagenicity [186,243,247], genomic instability [2,3,173,224,232,249,250,258,320], cancerous effects [225], and inflammatory effects [223] were at the center of the discussions, which was also because of the nonlinear responses that were incompatible with the LNT model. Later, closer insights revealed that bystander effects and NTE could also exhibit biopositive effects [29,70,72,107,130,191] covering all dose ranges, with systemic and longer distant abscopal reactions mediating biopositive effects by modulating innated and adaptive immune responses with favorable outcomes in anti-pain and anti-cancer RT [226,227,229,231,234,277], but, sometimes, also bionegative effects [228]. As shown in this paper, a form of red line is constituted by mitochondria involved in most IR responses. 

The following underlines that bystander and NTE mainly rely on the following: mitochondria-driven energy metabolism; mitochondria-dependent apoptotic signaling; mitochondrial reactivity to calcium fluxes; changes in mitochondrial membrane potential; mitochondrial ROS and NOS generation; mitochondrial DNA; mitochondria-dependent 53BP1 delocalization; cytokine and TGF-β release; mitochondria-dependent NF-κB/iNOS/NO and NF-κB/COX-2/prostaglandin E2 signaling pathways; the oxidative status of the bystander cells; the level of oxidative stress induced by IR of different radiation quality (LET), IR dose level; radiation-induced biophoton level (biophoton emission of irradiated cells); and IR-induced exosomes and their cargo (mtDNA, types RNAs, nDNA, cytokines, etc.). 

Table 1 indicates some important findings on the role of mitochondria in IR-induced bystander and NTE effects. Basically, this is due to their high energetic and metabolic reactivity, their very sensitive activity of signaling intracellular and extracellular insults as well as mediating cellular defenses (including potent apoptotic), but, also, their anti-oxidant and general innate and adaptive immunological responses [321].

Several reviews emphasize the involvement of mitochondrial functions in bystander effects and NTE [2,3,5,16,19,70,72,78,173,184,185,192,193,223,241,252,321,322] with exosomes [231,232,234]. 

**Table 1 ijms-24-11460-t001:** Role of mitochondria in low-dose IR nontargeted effects (NTE).

IR Exposure Exp. Device	Biological System	Observation	References
Low LET photons (γ-rays, X-rays) (0.5, 5 and 7.5 Gy) ICCM	Mammalian cells, Chinese hamster (CHO-K1) Human keratinocytes (HGV-G) Medium transfer	Absence of RIBE in CHO-K1 mutants with nonfunctional glucose-6-phosphate dehydrogenase (G6PD) involved in mitochondrial metabolism in HGV-G by inhibition of apoptosis and lactate metabolism. Alteration of calcium fluxes and loss of mitochondrial membrane potential (MMP). Involvement of mitochondrial ROS.	[318,319,323]
Low LET IR 5 mGy and 0.5 Gy ICCM	Human keratinocytes (HGV-G) Medium transfer	Reduction of clonogenicity. Induction of increases in mitochondrial mass and low Bcl-2 expression in bystander cells after 5 mGy in ICCM, but increased expression after 5 Gy. Decrease in survival.	[312]
X-rays 1Gy ICCM	Human hybrids cells: Chinese hamster ovary (CHO) cells GM10115 + human chromosome 4, Medium transfer	IR-induced mitochondrial dysfunction results in persistent high levels of ROS perpetuating genomic instability plus clastogenic and transgenerational effects.	[249,250]
IR: γ-rays 5 Gy ICCM	Chinese hamster (CHO-K1) Human keratinocytes (HGV-G) Medium transfer	Increase in mitochondrial mass, dysfunctional mitochondria in BE.	[324]
IR: γ-rays (5 mGy, 0.5 Gy, 5 Gy) ICCM	Chinese hamster (CHO-K1) Human keratinocytes (HGV-G) Medium transfer	Mitochondria are sensitive to LDIR and ICCM, loss of enzymatic functions (OXPHOS), and altered mtDNA-directed protein synthesis.	[325]
γ-rays or 160 kV X-rays (0.5 Gy)	Human mammary epithelial cells (HMEC); Balb/c mice TGF-β1 +/− and +/+	Increased centrosome deregulation as a function of time. After IR, clonal expansion CA increased in HMEC, but unstable cells could be deleted by TGF-β1 via p53-dependent apoptosis (involving mitochondrial signaling) TGF-β1 that can also suppress EMT.	[326]
Microbeam IR with Carbon ions or X-rays	Mammalian cells Murine lymphoma L5178Y in co-culture with irradiated neoplastic epithelial cells. Co-culture experiments	Cytoplasmic and cell irradiation affects mitochondria and calcium fluxes in targeted glioma and fibroblast cells. Cytoplasmic IR involved mitochondrial damage and RIBE response.	[244]
Microbeam with α−particles	T98G glioma cells and AG01522 fibroblasts ICCM or Co-culture experiments	Calcium signaling occurs early (RIBE). NO and mito-chondrial ROS lead to chromosomal damage (MN).	[243]
^241^AM source α-particles (100 mGy) ICCM	Hamster normal AL cells ρ+ and mtDNA-depletedAL cells (ρ0) (donor) and normal human fibroblasts (AG1522) (receptor cells). Medium transfer	Mitochondria-derived NO and O_2_^−^ play an important role in the initiation and activation of RIBE. IR-induced intracellular factors derived from mitochondria and calcium-dependent mitochondrial NOS. Mitochondria intercellular signaling from irradiated cells participates in ROS-mediated genotoxicity.	[327,328]
Microbeam IR with 1–10 protons ICCM	Human keratinocytes HGV-G Medium transfer	ROS levels increased in bystander cells. Apoptosis induced was associated with a decrease in MMP and increased intracellular Ca^2+^ levels.	[242]
Microbeam IR with α-particles	Cervical cancer cells (HeLa) and mitochondria depleted pseudo-ρ0 cells	No RIBE in the absence of mtDNA. Signaling is inhibited by ROS and RNA inhibitors. Mt-dependent 53BP1 delocalization. BE involves intact mt signaling from targeted cytoplasm to the nucleus.	[196]
Microbeam IR with 4He ions (120 keV/μm) α-rays	Human fibroblast cells ρ0 and ρ+	High BE mutagenic response in mtDNA depleted ρ0 cells. BE involved mt-dependent NF-κB/iNOS/NO and NF-κB/COX-2/prostaglandin E2 signaling and NOS and COX2 signaling.	[245]
Low-dose a-particles (0.29 mGy–25 mGy) and γ-rays (2 mGy–50 mGy	208F and v-src trans-formed 208Fsrc3 rat fibroblast cell lines. Co-culture experiments	Low-dose IR of non-transformed cells can induce apoptosis in precancerous cells through RIBE involving ROS/NOS signaling and cytokines, such as TGF-β. The stimulatory effect saturates at 50 mGy for γ-rays and at 25 mGy for α-particles.	[191]
1 GeV/u iron ions (LET~151 keV/μm), 600 MeV/u; silicon ions (LET~51 keV/μm), or 1 GeV protons (LET~0.2 keV/μm).	Normal human fibroblasts (AG1522) Test of progeny: co-cultures of cells exposed to low or high doses of high LET IR	RIBE depends on radiation quality and dose, and oxidative stress involving mitochondria.	[186]
γ-rays (0.05 and 0.5 Gy) ICCM	Human keratinocyte cell line (HaCaT) Medium transfer	Low-dose expression of genes involved in mitochondria-driven intrinsic apoptosis induced in bystander cells at low-dose (50 mGy).	[190]
γ-rays, α-particles and HZE particles (500 mGy)	Normal human FB (AG1522 cells) co-cultured with a-irradiated HeLa cells (500 mGy) (connexin 32) Co-culturing	Increased induction of MN and GI in bystander cells.	[224]
Tritium (β-radiation) induced UV biophoton emission	Human colon carcinoma cell line, HCT116 p53 +/+ Biophoton emission involvement in BE Exosomes	Biophoton electromagnetic bystander signaling compromises mitochondrial complex V (ATP production) and may be involved in the human fatigue syndrome. Exosomes extracted from UV-ICCM modulates clonogenic survival and MPP in bystander cells.	[38]
γ-rays (22 mGy) and biophoton emission	Cells: HCT116 p53 +/+ Test involvement of cellular emissions of biophotons in gamma radiation that is induced bystander cells	Low-dose biophoton emission from irradiated human cells may cause detrimental low-dose RIBE.	[39]
6 MeV photons (Clinac 600), 2 Gy	Fadu cells derived from HNSCC Secretion of exosomes in RIBE	NTE is propagated by mtDNA and RNA in vesicles similar to exosomes.	[28]
X-rays (0.1, 0.25, 2 Gy) Extracelluar vesicles (EVs)	C57BL/6 mice Total body IR, Extracellular vesicles (EVs)	A panel of miRNA are involved in EVs bystander effects, differently at low and high dose, IR induced systemic effects.	[329]
X-rays 4 Gy CCCM/ICCM	Seven-week-old male ICR mice: ELV from irradiated mouse serum and ICCM	Absence of DNA damage in CCCM ELV or ICCM ELV from mt-depleted. ρ0 normal human fibroblasts. Secretion of mtDNA via exosomes is involved in mediating RIBE signals.	[32]
X-Rays Partial and whole body exposure 2 Gy	C57Bl/6 female mice of eight weeks of age Analysis of ‘Out of field ‘effects partial body IR in mice Exosomes	Deregulation of many proteins and miRNAs. Some miRNA, proteomic changes, and exosomes are involved in anti-apoptotic effects. Injection of exosomes from irradiated mice can prevent apoptosis.	[226,227]
γ-rays, high doses (2–8 Gy) ICCM	Human HepT2 cells Medium transfer from irradiated cells	Induction of Bax, Bcl2, caspases and γ-H2AX DNA damage in bystander HepT2 cells.	[330]
200 kV X–rays (6 Gy)	Human pancreatic cancer cells (MiaPaCa–2), wild–type (wt) and ATM−/− fibroblasts Co-culturing	Healthy ATM+/+ cells modify the DDR of irradiated cells by a microtubule- and ATM-dependent exchange of healthy mitochondria.	[201]

In IR-targeted cells, the availability of intercellular gap junctions, micronanotubes (MNTs), and the biophoton emission of irradiated cells can be implicated in the initiation of bystander effects (see review [331]). Apparently, fast growing LD-resistant cell lines (HT29, PC3) were more reluctant to produce bystander effects than slower growing radiosensitive cell lines (HaCaT or SW48). Repair deficient cell lines gave stronger bystander signals than DNA repair competent cells. Exosomes from cells that received biophotonic UVA signals from irradiated cells may be able to produce bystander effects in unirradiated cells [38]. Such UVA exposures from stressed cells may cause ATP depletion, explaining fatigue, decreased DNA repair, and immune activity [130].

As already shown above, several lines of evidence also indicate the involvement of mitochondrial functions in bystander-induced genomic instability [2,3,23,173,224,232,250,252,254,255,258,320,332], in systemic abscopal effects [29,225,229] and in IR-induced immune effects [193,204,223].

NTE is essentially a low-dose effect that is triggered by acute exposures as low as 2–3 mGy, and it increases until it saturates at about 0.5 Gy (in vitro) [333].

Among the factors playing a role in RIBE and NTE, calcium is an important secondary messenger released from the endoplasmic reticulum (ER) affecting both mitochondrial functions and bystander responses in co-culture experiments [243,334].

Mitochondrial ROS can promote cytokine upregulation (IL-6, IL-8) and their release into the growth medium. After IR, IL-6, IL-8 TNFα, and IL-33 are released in a NF-κB-dependent way. The medium stimulates NF-κB and MAPK pathways and increases AKT activity, and IR-induced NO can induce the release of TGF-β3 from irradiated cells and is a signaling factor for free radical induced DNA damage in bystander cells, decreasing cell viability [335]. 

RIBE and NTE depend on the P53 status, the mitochondria-dependent energetic and physiological state, and the oxidative status and the physiological state of the cells as well as the organism [326]. 

The dynamics of mitochondria and their plasticity with fission, fusion, and change in copy numbers and DNA content, controlled by autophagy and mitophagy, make mitochondria very flexible for exchanges between cells and cancer cells, surrounding normal and immune cells in the environment. 

Bystander effects are double-edged swords [224] given that they can have protective and toxic effects communicated through intercellular gap-junctions involving connexin expression Cx26 or Cx43 in co-culture, die or undergo proliferative arrest, or connexin Cx32 expressing DNA damage in later passages (genomic instability).

Low doses of IR have been shown to have both damaging and beneficial effects, including bystander and NTEs [72]. Damaging effects include inflammatory effects, cell inactivation, premature senescence and aging [336], mutation and cell transformation, and cancer induction [193]. 

### 10.2. Factors Possibly Contributing Adaptive Beneficial and Armful Effects of Low Doses

In recent years, it has become evident that low-dose effects of IR(LDIR) can be adaptively beneficial as well as detrimental. Often, nonlinear effects have been observed at low-dose exposures in contrast to high-dose exposures for important biological endpoints [4]. In fact, they are often beneficial, and are more rarely harmful [66,92,337]. These possible dichotomic effects are less apparent at high doses, where harmful effects follow a linear induction pattern, and where the LNT model is dominant. 

The repair of DNA DSBs was not linear in the very low-dose range. Moreover, DSBs induced by a very low dose (1 mGy) of X-rays were efficiently repaired in proliferating normal human fibroblasts but not in quiescent fibroblasts [77].

Genetic factors involved in beneficial hormetic responses and harmful responses partially overlap [70], and LDIR-induced hormesis was shown to include ATM, extracellular signal-related kinase (ERK), mitogen-activated protein kinase (MAPK), phospho-c-Jun NH(2)-terminal kinase (JNK), and protein 53 (P53)-related signal transduction pathways. MAPK and p53 are also involved in adaptive responses. On the other hand, LDIR-induced bystander effects and genomic instability may include COX-2, ERK, MAPK, ROS, tumor necrosis factor receptor alpha (TNFα), and ATM, ERK, MAPK, P53, ROS, and TNFα-related signal transduction pathways, respectively.

### 10.3. LDIR Adaptive Responses

Although LDIR effects are still controversial, without any doubt, LDIR can induce important adaptive responses in mammalian cells and in animals [338]. Park et al. revealed that chronic low-dose IR (LDIR) (10 and 50 mGy) and a challenging dose of 2 or 10 Gy resulted in increases of AKT, acinus protein via NF-κB activation in different human normal, and tumor cells [339]. Clear differences in normal cell and tumor cell responses to LD and HD were observed, which likely depend on AKT activation regulated by protein phosphatase 2 (PPA2). LD chronic exposure of normal cells lacking basal Akt activity increased activation of the ERK pathway involved in adaptive IR responses.

Regulators such as cyclin D1/CDK4 and cyclin B1/cyclin-dependent kinase 1 (CDK1) complexes are mediators between IR-induced DNA damage and mitochondrial functions regulated through phosphorylation of mitochondrial targets. They can lower genotoxic stress by adjusting mitochondrial metabolism, and they can enhance cellular homeostasis. The cell cycle and cell cycle-associated proteins are regulated by a dose of 200 mGy. High doses may give rise to p53 independent signaling and inhibit apoptosis and cell cycle progression via p21 phosphorylation, whereas LDIR may activate p21, which can inhibit CDK1, CDK2, and CDK4/6, allowing cell cycle progression, IR-induced hormesis, and adaptive responses. LDIR can also activate p53, p21, and apoptosis, thereby inhibiting oncogenesis [338]. In this view, hormesis can also be achieved by enhancing DNA repair, ROS/RNS production, activating Nrf2 and NF-κB, and increasing antioxidant defenses. Moreover, cell proliferation may be enhanced through activation of signaling pathways (PI3K/AKT, Ras/Raf/ERK, and Wnt/b/β-catenin), and the innate and adaptive immune system can be activated by stimulating cytokine production. Guéguen et al. demonstrated that LDIR elicited DNA damage repair pathways (involving p53, ATM, and PARP), the antioxidant pathway Nrf2, and the immune inflammatory response (NF-κB pathway), cell survival/death pathway (apoptosis), the endoplasmic response to stress (UPR response), and other cytoprotective processes, including autophagy and cell cycle regulation [148].

miRNAs also play a role in LDIR responses. Wang et al. observed that 50 mGy adaptive apoptosis prior to 20 Gy in A549 lung cancer cells, 16 miRNAs were differently expressed and involved in this LDIR response [340]. 

After low-dose exposure, human cell nuclear retention of cyclin D1 plays an important role in mitochondrial ROS mediated genomic instability [173]. Mechanistically, mitochondrial ROS perturb AKT-cyclin D1 cell cycle signaling through oxidation of PP2A, leading to the accumulation of nuclear cyclin D1 and genomic instability [341]. 

LDIR of 20–500 mGy induced clusterin, a survival protein, which is involved in adaptive responses and radioresistance in cultured human cells and in mice in vivo. This is secreted after LDIR, and it is probably involved in the development of genomic instability as well as the modification of intracellular communication by binding to cell surface receptors (TGF-β receptors) [342].

Ahmed et al. provided evidence for cooperative functions of ATM, ERK, and NF-κB in inducing a survival advantage in human keratinocytes through a radioadaptive response after LDIR treatment (100 mGy) with X-rays) [343].

Low-dose HRS was observed in radiation-induced acute myeloid leukemia. HRS stimulated cell killing and Sfpi1 deletions, thus enhancing the cancer risk by altering the probability of Sfpi1 deletions to both occur and persist [344]. HRS induction in rAML cells at low doses (60 mGy) involved oxidative stress and an increase in ROS in these hematopoietic cells [345].

As seen in the work of Kabilan et al., LDIR can orchestrate hormesis, including the factors p53, NRF2, signaling pathways ATM/ERK/NF-κB, PKC-p38MAPK-PLC and AKT/ERK/TNFα, FOXO3A, and TGF-β, as shown at the transcriptional, translational, and post-translational levels [150]. The physiological outcome depends on the balance between sustained damage and the DDR signals. This persistent adaptive state may then trigger genomic stability and enhanced immune functions, contributing to longevity and protection from cancer. These authors also reported that the sensing and repair of DSBs was altered by lowering the translation factor elF4G1 targeting translation of BCRCA1 after IR. BCRA1 involved in error-free homologous recombination repair was shown to play a role in regulating transcripts of genes involved in DDR and DDR-signaling [150]. It was able to shift the balance of error-prone NHEJ towards error-free HR pathway of DSBs, which can be regulated by 53BP1 and through elFG1 translation control. The authors suggested that translational reprogramming of DSB signaling and repair is part of LDIR responses that may lead to radiation hormesis associated with lower neoplastic transformation, suppression of tumorigenesis, and extended lifespan.

As shown by Fernando et al., the effects of low-dose radium alpha rays may be quite different in different species. Human keratinocytes showed radioresistance whereas the embryonic Chinook salmon cell line (CHSE-214) was resistant to γ-irradiation but exhibited radiosensitivity towards exposure to alpha particles [346].

Proteomic analysis of the bystander effects induced in chondrocytes by chondrosarcoma cells exposed to X-rays and C-ions at 100 mGy revealed about 20 proteins that are involved in oxidative stress responses (mitochondria), cellular motility, and exosomes pathways [347]. The conditioned medium contained 40 modified proteins. In a low-dose (100 mGy) condition, DNA damage-responsive genes were increased in chondrocytes and, also, in a large cluster of proteins involved in stress granules with likely cell protective functions. Some bystander effectors showed specificity in terms of radiation quality, i.e., towards X-rays or carbon ions. Meanwhile, translational proteins were associated with both, antioxidant pathways and IL-12 were associated with X-rays, and G1/S and G2 DNA damage were specific to C-ion exposure [150]. 

### 10.4. Immune and Anti-Tumor LDIR Effects

Concerning immunological effects, LDIR induces possible anti-inflammatory effects at low dose, expression of pro-inflammatory cytokines at moderate doses, and immunosuppression after higher doses (precursor cell death as well as exacerbated innate immune responses [348]. ATM can trigger NF-κB activation together with nucleoplasmic shuttling involving the NF-κB essential modulator. The activation of NF-κB involves the release of the complex with IκB kinase. Cytokine expression can activate NF-κB, and NF-κB can also be activated via TLRs (Toll receptors) by danger signals from dying cells. Activation of the immune response can be beneficial or harmful (detrimental). 

At LDIR, possible anti-cancerogenic effects can be observed. In RT, anti-tumor responses may be supported by TLR agonists activation of NF-κB. Interestingly, LDIR at 50 mGy could activate NF-κB on two phosphorylation sites (Ser36 and Ser418) without inducing genomic instability, probably due to efficient DNA repair [349].

Whole-body LDIR on metastatic mouse models induced anti-tumor responses via alterations of the immunosuppressive tumor environment, leading to a reduction of pro-inflammatory Ly6chigh monocytes in APOE−/−mice [350,351]. LDIR was upregulating selected immune components INF-γ, IL-4, and Il-5, and cytokines released stimulated CD4+ cell T-cells [274]. Fractionated LDIR also caused NF-κB upregulation.

Moreover, tissue specificity also plays an important role in LDIR effects. For example, low doses resulted in neurological effects in exposed individuals [352]. Indeed, cells are very sensitively reacting to low-dose insults, but the reactivity depends on the energetic and physiological state of the cell and also on the presence of concomitant insults from environmental factors.

In recent years, furthermore, stress granules have become a topic in cancer research [353]. They are involved in various tumor-associated signaling pathways, including cell proliferation, apoptosis, invasion and metastasis, chemotherapy resistance, radiotherapy resistance, immune escape, and bystander effects [354], and this, also, at low doses [347]. Interestingly, in HNSCC cells, ceramide-enriched membrane domains contributed to targeted and nontargeted effects of radiation through modulation of PI3K/AKT signaling [355]. In radiosensitive SCC61 cells, NTE effects were brought about by the formation of an IR-induced ceramide-enriched domain. In radioresistant SQ20B cells, such domain allowed phosphatidylinositol-3-kinase (PI3K)/AKT signaling. Disruption of membrane lipid rafts led to the radiosensitization of these cells.

Thus, intracellular stress granules should also be considered in radiation research [356] because they are linked to the general stress-responsive intracellular and intercellular network, including mitochondria [357].

### 10.5. Biopositive Effects of LDIR

Moreover, mitochondrial functions can be activated by LDIR boosting mitochondria-dependent immune reactions and activation of Nrf2 [103]. Such immune cell activation concerns certain blood cells (lymphocytes and eosinophils) [104] as well as erythrocytes [69].

Mobilization of cellular defenses (antioxidants) as well as the recovery of signaling systems from cognitive and intellectual deficiencies were found in the case of Alzheimer’s disease [100,101,102].

### 10.6. Biopositive Effects of LDIR on the Immune System

NK cells can be activated by LDIR in mice [358]. LDIR by 75 mGy of X-rays upregulated the Th1 cytokines IL-1β, IL-2, IFN-γ, and TNF-α, and it downregulated IL-10 pro-duction on days 12, 16, and 20, whereas HDIR can inhibit the production of these cytokines. LDIR pretreatment protected the cytokine-producing ability of splenocytes on days 12, 16, and 20 to some degree, but this effect did not last up to day 24. LDIR induced NK cell activation, also in vitro, most likely through the p38 mitogen-activated protein kinase pathway, the molecular mechanism of LDIR-induced antitumor immunity enhancement. Interestingly, regulation of the Akt and the 38 pathway could alleviate Alzheimer’s disease in drosophila [98].

LDIR (0.1 or 0.2 Gy of X-rays) can inhibit metastases and trigger the cytolytic NK activity in Balb/c mice. However, the mechanism involved is not yet clear [114]. 

After repeated low-level 10 daily exposures of radiosensitive BALB/c or radioresistant C57BL/6 mice to 10, 20, and 100 mGy of X-rays, NK cell-enriched splenocytes obtained from the animals showed significant up-regulation of their anti-tumor cytotoxic function. Peritoneal macrophages also exhibited cytotoxic effects on tumor cells and increased NO production [359].

Following human peripheral blood exposures to LDIR of 50 and 150 mGy, transcriptomic profiling of gene expression showed upregulation of many genes, such as HLA-DQA1, HLA-DQA2, HLA-DQB2, HLA-DRB1, and HLA-DRB5 involved in antigen processing and presentation, immune system-related diseases, and cytokine-mediated signaling [360]. This suggested that the immune system had been boosted. Positive immune-stimulatory responses were also found in isolated human primary monocytes with the activation of toll-like receptors (TLRs), mitogen-activated protein kinases (MAPKs), and NF-κB signaling, especially after LDIR exposure to low doses (0.05 and 0.1 Gy). P53 was not involved.

The beneficial effects of LDIR in combination with immunotherapy were also observed by Barsoumian et al., who showed that LDIR enhanced systemic antitumor responses by overcoming the inhibitory stroma in established 344SQ lung adenocarcinoma in 129Sv/Ev mice [361].

### 10.7. Bionegative Effects of LDIR

In mice, LDIR can be protective [91], but higher doses can be detrimental [91]. LDIR can affect neurologic functions by downregulating neural pathways [99].

LDIR (γ-rays) may program macrophage differentiation to an NOS (+)/M1 phenotype that orchestrates effective T-cell immunotherapy [276], and the tumor environment is modified by retuning tumor-associated macrophages [362].

Recently, conditioned medium from irradiated WI-38 lung fibroblasts and H1299 lung adenocarcinoma cells exposed to 0.1–1 Gy enhanced the migration and the invasion of unirradiated H1299 cells without inducing apoptosis but senescence in a c-Myc-dependent way [363]. This suggests that the bystander responses can be dependent on the particular oncogenic state of the cells.

### 10.8. Low Dose-Rate Effects

Low-dose rate (LDR) effects play an important role in the low-dose effects of IR. As recalled by Amundson et al., classical observations on protracted low-dose radiation exposures yield normal DNA repair competent cells important sparing effects due to efficient DNA repair [364]. LDIR go often together with LDR exposures (see radioadaptive responses). LDR exposures are generally protective against mutation induction and cell transformation in vivo and in vitro. As shown by Rothkamm and Löbrich, at a very low-dose 1 mGy of IR on proliferating cells, high-dose rates rapidly turn on the DDR pathway involving ATM and phosphorylation of H2AX (γH2AX) [77]. In resting cells, this was not the case, and DNA repair was stalled at (1–20 mGy). Collis et al. were able to show in cancer cell lines that low-dose rate IR (i.e., 450 times lower than a high-dose rate producing environ 4–5 DSB/h instead of 1800 DSB/h) increased cell killing (clonogenicity) as a consequence of inefficient activation of the DNA damage sensor ATM and H2AX phosphorylation [365]. Thus, DDR signaling is nonlinear at LDR, even though inverse dose-rate effects have been observed with increased mutagenicity in somatic and germ cells due to cell cycle dependent radiation sensitivity windows. DNA repair deficient cells, for example. Fibroblasts from AT patients deficient in ATM exhibit little or no dose-rate effects. Gene expression studies revealed that low-dose rate exposures triggered protection against protection against the induction of apoptosis with a linear induction o p53 regulated genes, except MDM2 and genes regulated by cell cycle.

It is crucial to note that dose rate is an important factor in modulating mitochondrial biogenesis [14]. A low dose of 0.1 Gy at a dose-rate of 0.055 Gy/min caused dysfunction of the mitochondrial respiratory chain in rat small intestine enterocytes, with perturbance of cytochromes in the inner mitochondrial membrane and inhibition of H^+^—ATPase activity [366].

Barjaktarovic et al. have shown that whole body IR of ApoE−/− mice (deficient in cardiac mitochondrial protein (associated with metabolic impairment and sirtuin downregulation) at chronic exposure at 20 mGy/day for 300 days induced increased acetylation and reduced mitochondrial sirtuin [367]. The hyperacetylation involved the mitochondrial TCA cycle, fatty acid oxidation, oxidative stress, and the sirtuin pathway. Acetyl-CoA increased, and cardiac metabolic regulators (PGC-1 alpha and PPAR) were inactivated. 

Interestingly, LDR exposures of normal human cells (48BR) induced activation of mitochondria-dependent oxidative stress involving AMPK, p38, MAPK, and ERK, but this was not seen in cancer cells [26]. 

LDR exposures are important in adaptive responses. For example, Sugihara et al. found that a low priming dose at a low-dose rate (20 mGy/day) allowed an adaptive response 12 days later to a challenging dose of 6.75 Gy at a high-dose rate [368].

Several excellent recent reviews [369,370,371] give a detailed account of the key events of LDR effects in animal models. Prolonged life times of mice after chronic low-dose rate IR exposure could be observed [372,373]. However, Ogura et al. reported an increased copy number variation (deletions) in the offspring of male mice exposed to LDRIR associated with a possibly shorter life span [374]. Braga-Tanake et al. also reported that chronic 1 mGy/day exposure of mice yielded significant changes in lifespan, neoplasm incidence, chromosome abnormalities, and gene expression [375]. In fish cells, a low-dose rate of 83 mGy/min exposure involved a biphasic response, leading to a higher clonogenicity than after high-dose rate 366 mGy/min [376]. Nonlinear responses to low-dose rate exposures were observed in the epithelial cells of the lens [377] and also in human umbilical endothelial cells [378,379]. 

Immunological IR responses are also affected by LDRIR. For example, Ina et al. showed that exposure of wildtype mice to chronic radiation 1.2 mGy/h increased CD4+T cells and CD8 molecule expression, while CD40+ B cells decreased [380]. Chronic exposure at LDR activated the immune system of the whole body. They also reported that IR-induced lymphoma at a high-dose rate was suppressed by pretreatment with low-dose radiation at 75 mGy (adaptive response) and further repressed by lifelong γ-IR at LDR 1.2 mGy/h (which on its own did not yield lymphomas or other tumors) [380]. 

Rey et al. have shown a decrease in proinflammatory Ly6CH monocytes [351], and Edin et al. (2015) have shown an activation of TGF-β3 at LDRIR [381].

LDR exposure (5 mGy/min) for 1 h induced protection against lethal IR dose effects without affecting the lifespan of DBA/2 mice [382]. In human fibroblasts, genes against oxidative stress were upregulated at a low-dose rate [383].

Recent findings reveal that the main target of chronic ionizing radiation is the activation of the inflammatory system, which can lead to the initiation of related processes, such as apoptosis, cell differentiation, and proliferation, angiogenesis, invasion, and metastasis in tumor progression [384]. 

In addition, dose rate and dose fractionation and FLASH exposures [385,386,387] determine the biological consequences and outcomes of IR. Interestingly, FLASH with protons (FLASH-RT) prevented mitochondria damage characterized by morphological changes, functional changes (membrane potential, mtDNA copy number, and oxidative enzyme levels) and oxyradical production [387]. The Dynamin-1-like (Drp1) protein mediated mitochondrial homeostasis in FLASH-RT [385,386,387].

This also confirms that not only radiation dose but also radiation dose rate and radiation quality are all important for IR-induced biological responses and RT. In fact, low-dose and low-dose rate IR effects have opened up many new avenues in radiation biology that will be very beneficial for both recognizing and understanding important networks involved in LDIR and LDRIR responses, which will pave the way for better radiation protection and anti-cancer radiation therapies. 

### 10.9. Relationship of LDIR and LDRIR to DDR and Mitochondrial ROS

The activation of ATM not only includes DDR but also ROS-sensing, apoptosis, and senescence [262]. After the induction of metabolic stress, ATM is not only acting in the nucleus to cope with DNA damage but also interacting with organelles and molecules in the cytoplasm [258]. Mitochondria are known to function at the crossroads of ATM mediated stress signaling and regulation of cellular ROS, but ATM can also modulate mitochondrial gene expression [259]. ATM activated by oxidation promotes the formation of active covalent ATM-ATM dimers independently of MRN and DNA. This covalent dimer (via disulfide bonds) regulates cellular ROS, mitophagy, homeostasis of proteins, and ROS-dependent autophagy [259]. If activated, ATM may inhibit apoptosis but promote senescence. Absence of ATM functions causes cerebellar degeneration and genomic instability [256,257]. Very importantly, a fraction of ATM is localized in mitochondria, and is thus participating in the general signaling platform of mitochondria and activated by mitochondrial dysfunction [260,261]. On the other hand, ATM is critical for the control of cellular redox homeostasis [255], which, if perturbed, may result in cancer through elevated mitochondrial ROS production mediating genomic instability, chronic inflammation, and the development of an active tumor microenvironment. It should be noted that genomic instability could be induced by low-dose IR in peripheral blood lymphocytes [253]. Excessive levels of mitochondrial ROS appears to interfere with AKT/cyclin D1 cell cycle signaling via oxidative inactivation of protein phosphatase 2A after low-dose long-term fractionated IR [173]. cGAS surveillance of MN links genomic instability to innate immunity [305]. IR-induced double-stranded DNA fragments from MN are cognized as an important source for immune-stimulation.

The different chapters of this paper provide several lines of evidence that IR responses do not follow the same reaction schemes depending on low and high dose, dose-rates and dose fractionation, and radiation quality. The differences in short-term and long-term IR responses are especially evident regarding the responses in terms of initial cellular damage and the intracellular and intercellular signaling that are induced, including mitochondria and mitochondrial functions. Tight links between initial damage, NTE bystander effects, and immune effects exist involving mitochondria linked to biological consequences. Mitochondria play an important role in the management of IR-induced initial damage in its evolution post-radiation with NTE bystander and with immune effects. Thus, the mitochondria are important determinants of the biological consequences of IR, which provide a new understanding of the differences in efficacy of low- and high-dose IR in radiation protection and antitumor RT.

### 10.10. Role of Mitochondria in Radioresistance

As already mentioned above, recognition of foreign particles and molecules in the cells is essential for cellular defense strategies [17,388]. For these, mitochondria have a central and prominent position. They are involved in cellular signaling of all types of damages, they provide the energy for the different types of cellular responses, they direct innate and adaptive immune responses and coordinate long term responses, and they play a pivotal role by determining final adverse or beneficial outcomes.

Depending on the characteristic nature and type, dose, and dose rate, IR is able to elicit, in some particular circumstances, damaging or beneficial outcomes involving specific mitochondrial reactivity. To some extent, ATM is also involved.

However, the induction of radioresistance involving ATM, cyclooxygenase-2 (COX-2), ERK, JNK, reactive oxygen species (ROS), and P53 is somewhat ambiguous: in normal cells, radioresistance is a biopositive effect, whereas in antitumor RT, it is considered to be a bionegative effect and a serious drawback counteracting RT efficiency.

Since mitochondria constitute the main cellular power station, it is not surprising that mitochondria are involved in radioresistance [389]. It has been known for some time that radioresistance of cells involves mitochondrial glucose metabolism, including glycolysis and oxidative phosphorylation [390]. Malignant transformation, tumor progression, and evasion of exogenous stress are also influenced by mitochondria metabolism [391,392].

Warburg has already found that cancer cells can undergo aerobic glycolysis with increased glucose uptakes, glycolysis, and high lactic acid production [393]. 

Cancer cells are characterized by the mitochondrial synthesis of NADPH through the pentose phosphate pathway and the decrease in oxidative phosphorylation and the dependence of tumors on glycolysis [394]. In the acidic condition of tumor environments, cancer cells are able to reduce extracellular acidification and increase O_2_^−^ production by switching from glycolysis to oxidative phosphorylation [390], thereby promoting tumor invasion and radioresistance to RT [395]. In fact, glycolysis is upregulated in most tumors without mitochondrial dysfunction. In these cancers, OXPHOS continues normally, even producing as much ATP as normal tissue at the same partial pressure of oxygen [396]. The kinase AKT can interfere with mitochondrial metabolism, enhance aerobic glycolysis and mediate radioresistance in human tumors [397].

Many inhibiting molecules have been developed against ROS and oncometabolites or to regulate OXPHOS and apoptosis, which can target specific receptors and enhance radiosensitization of tumor tissue. However, some are lacking specificity, and they have to be adjusted individually to the tumor type to overcome radioresistance [223]. 

Numerous reports show that metabolic inhibitors can interfere with mitochondrial metabolism and confer radiosensitization effects (see [314]). For example, inhibitors such as 2-deoxy-D-glucose of mitochondrial glucose metabolism radiosensitize cancer cells [398]. 

Oxidative phosphorylation inhibitors that affect mitochondrial function and reverse radioresistance [314] include metformin and phenformin affecting complex 1 as well as cervical cancer, head and neck squamous cell carcinoma, glioblastoma (IDH-wildtype), breast cancer, arsenic trioxide (As_2_O_3_), the treatment of acute promyelocytic leukemia and recently radioresistant solid tumors, and some cancer cells of the lung and liver. Atovaquone, an inhibitor of electron transport complex III, also significantly increased oxygenation and sensitized tumors to radiotherapy and radiosensitized hypopharyngeal, colorectal, and lung cancer cell lines. High-grade radioresistant gliomas can be radiosensitized by dichloroacetate through activation of OXPHOS by reversing aerobic glycolysis [399]. Radioresistant cervical cancers were shown to be sensitive to the inhibition of glycolysis and redox metabolism [400].

RT itself affects mitochondrial energy metabolism, mitochondrial morphology, and functions, and mitochondrial DNA mutation rates, respiration, and ATP levels are increased. Changes in mitochondria membrane potential and in mitochondrial energy metabolism are primary events in tumorigenesis and radioresistance in RT.

Factors that affect MMP and confer radiosensitivity are, to take a few examples, growth differentiation factor-15 (GDF15) belonging the TGF-β superfamily, which may represent a target to radiosensitize head and neck cancer cells by reducing MMP activation and allowing ROS generation; inhibitors of MEK/ERK-mediated signaling, such as PD98059, which increase MMP and FAS-mediated cell death (and caspase-8 activity); inhibitors of histone deacetylase reduce MMP and increase ROS generation together G2/M phase cell cycle arrest with apoptosis in esophageal cancers. Apparently, the blockage of the mitochondrial potassium -ATP (KATP) channel and ROS-induced MAPK/ERK kinase activation can radiosensitize glioblastomas [401]. For example, ROS generation and apoptosis could be enhanced in squamous cell carcinoma by poly-drug elevation of ceramide levels, and radioresistance could be overcome [402].

Concerning ROS generation, supplementation of cancer patients with antioxidants can be detrimental with adequate antioxidant status (lung, gastrointestinal tract, head and neck, and esophagus) but beneficial to individual cancer patients with deficient antioxidant systems [403].

Among the list of inhibitors of mitochondrial metabolic functions [314] are inhibitors of glucose transporters, agents increasing ROS and oxidative stress, agents altering MMP, up-regulators of pro-apoptotic genes (BAC and BAX), and inhibitors of NF-κB (for example curcumin). If ROS levels continue to increase beyond the antioxidant capacity of cells, this will cause apoptosis, ferroptosis, or cuproptosis, and it will significantly improve the efficacy of radiotherapy [392].

Zaffaroni et al. revealed other promising inhibitors that can increase the effectiveness of RT against cancers (breast, brain, melanoma, prostate, and ovary) [404]. One promising drug is lonidamine, an inhibitor of aerobic glycolysis in cancer cells. It affects the succinate-ubiquinone reductase activity of mitochondrial complex II, leading to enhanced ROS. Thus, glycolytic/mitochondrial metabolic changes appear to mediate cellular radioresistance.

A change in epigenetic regulation appears to be another important factor in radioresistance. Compared to nDNA, mtDNA is mostly hypomethylated. Epigenetic regulation is based on the mitochondria-specific DNA methyltransferase (mtDNMT). Under IR-induced oxidative stress, CpG islets in mtDNA are oxidized and are not available as methylation sites, and mtDNMT is inhibited. This affects regulation gene expression and genome integrity. Moreover, it may interfere with the production of important co-factors, such as, for example, ATP and acetyl-CoA involved in the acetylation of histones [6].

Due to the important role of mitochondria in metabolism and cell death [405], many factors are still to be discovered. One is papaverine, an inhibitor of mitochondrial complex I, which causes increased radiosensitization of solid tumors via oxygenation without important side effects in RT [406]. Another is pyrazinib, which is radiosensitizing radioresistant oesophageal adenocarcinoma via the modulation of mitochondrial bioenergetics [407]. 

Undoubtedly, mitochondria play a role in radioresistance, and adaptive responses of IR were observed in some tumors. For example, Aravindan et al. showed that LDIR can induce an adaptive response via the activation of NF-κB dependent responsive tumor necrosis factor α (TNF-α), interleukin 1a, cMYC, and SOD2 via intercellular communication and sequential orchestration, endorsing radiation protection (radioresistance) of surviving tumor cells [408].

Radioadaptive resistance of glioblastoma in RT involved the heme-containing enzyme cytochrome C oxidase affecting the cellular iron pool and IR-induced Fenton reactions with hydroxyl radical production [409]. Interestingly, a disruption of mitochondria was shown to radiosensitized prostate cancer cell lines [410]. Clearly, radioresistance of tumor cells can be reversed by targeting mitochondrial metabolism [314]. 

The direct implication of mitochondria in radioresistance has also been demonstrated by Grasso et al. [411]. They compared a radiosensitive SQD9 wildtype clone of HNSCC cells with a radioresistant SQD9 derived clone that harbored about 50% more mitochondria, a denser network around the cell nucleus, and about 35 more mtDNA. Apparently, mitochondria protected against IR-induced damage. Thus, the targeting of mitochondrial metabolism remains a valuable option in anticancer RT [412].

Furthermore, radiation quality also determines how strongly RT can modulate mitochondrial functions and whether mitochondria can even mitigate long-term radiation injury [19]. 

High LET IR (for example, Carbon ion therapy (CIRT)) is locally very damaging for mitochondria, particularly in tumor cells, but it leaves the tumor microenvironment and the surrounding immune cells largely intact in order to mobilize immune anti-tumor responses via dendritic and CD8+ T cells. Carbon ions strongly induce complete apoptosis and the death of tumor cells as well as immunogenic tumor cell death. Instead, low LET IR appears to affect the tumor environment, too, together with the surrounding mitochondrial energy supply for surrounding immune cells. It follows that low LET IR is less immunogenic for tumors and needs some additional support through immunotherapeutic means (gene-mediated immune therapy, vaccine therapy, immune checkpoint inhibitors (anti PD1, anti-PD-L1), etc. [30].

### 10.11. Conclusive Thoughts

Regarding the important role and the complexity of IR-induced mitochondrial interactions with the cellular network of pathways, it is now time to adopt broader views in radiation biology. In the near future, systems biology and artificial intelligence need to be considered as approaches to assist researchers to obtain a better understanding of the complex molecular biological networks involved in LDIR responses.

From this, it is now evident that radiation research has now reached another dimension of comprehension, promoting a holistic view of the mechanisms governing integrated cell and tissue responses caused by external insults [19]. In fact, IR has brought to light cellular defense systems and new aspects of their intimate connectivity. Different periods of research may be distinguished. This started with the determination IR effects and, especially, the deciphering (decipherment) of DNA repair, and has been followed in more recent years by the elucidation of wide-ranging signaling and the deciphering of immunological processes that are induced. Apparently, low-dose IR leads to efficient signaling, eliciting the cGAS-STING cascade and immunogenic responses. Nonlinear responses exist, especially in the low-dose range, indicating the complexity of interacting molecules and pathways. On the other hand, high-dose IR and RT gives rise to immune responses that start with the emission of immune-stimulating factors from dying cancer cells, which elicit impressive immunogenic potentials. With this, radiation research has over the years provided new insights into formerly hidden cellular networks that constitute the astonishing complexity and incredible refinement of biological systems. From a practical point of view, there is hope that the accumulating radiobiological knowledge will allow a better understanding of the mechanisms involved in radiation responses, and that it will help to define, more rationally and more precisely, beneficial and adverse outcomes of IR on humans and their environment. 

## Figures and Tables

**Figure 1 ijms-24-11460-f001:**
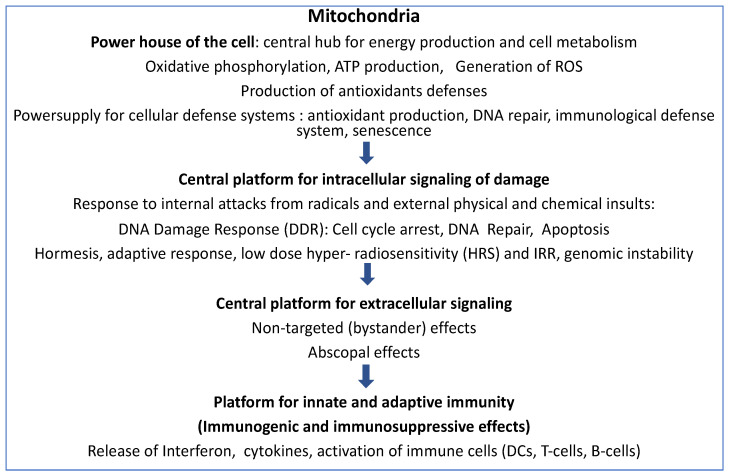
Mitochondria and IR responses: Mitochondria are the essential platforms of energy production and cell metabolism, cellular defense systems, intra-and extracellular signalization, and communication, as well as for innate and adaptive immune responses, especially after exposure to ionizing irradiation.

**Figure 2 ijms-24-11460-f002:**
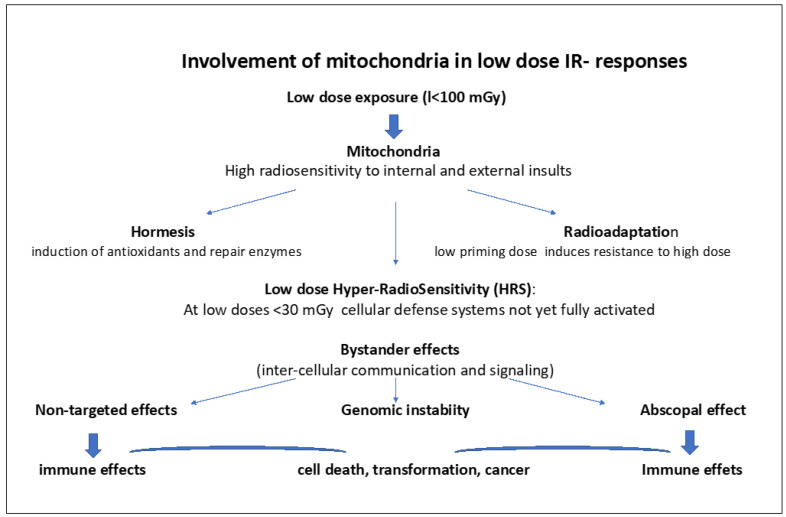
Involvement of mitochondria in low dose IR-responses.

**Figure 3 ijms-24-11460-f003:**
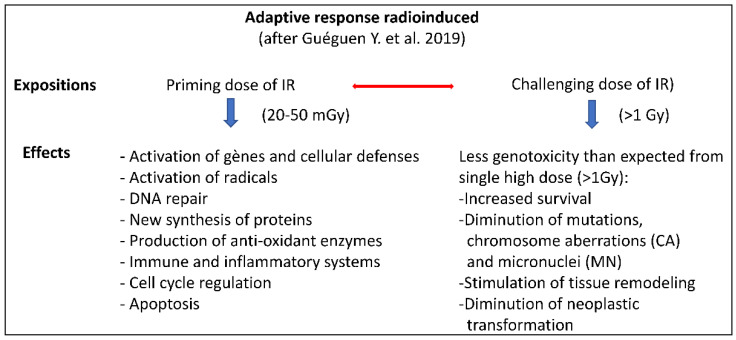
Scheme of the adaptive response radioinduced together with the biological consequences of the initial low dose priming those following the high challenging dose in comparison to the effects of high-dose exposure alone, [148].

**Figure 4 ijms-24-11460-f004:**
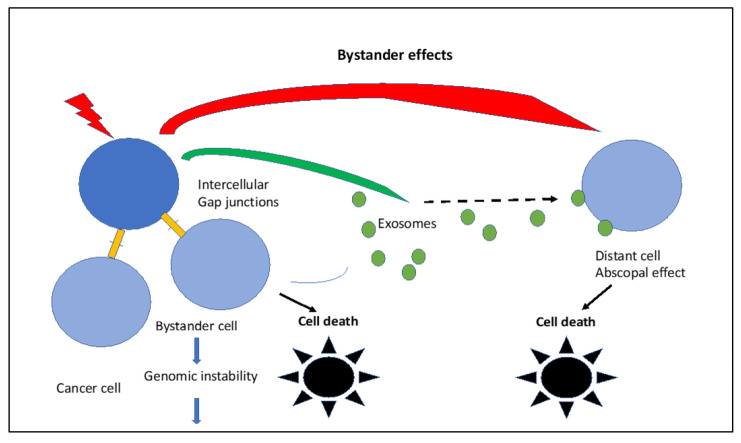
IR-induced bystander effects via intercellular gap junctions to neighboring cells and/or to distant cells via exosomes. Both are mediated by mitochondria and can lead to cell death, especially in cancer cells.

**Figure 5 ijms-24-11460-f005:**
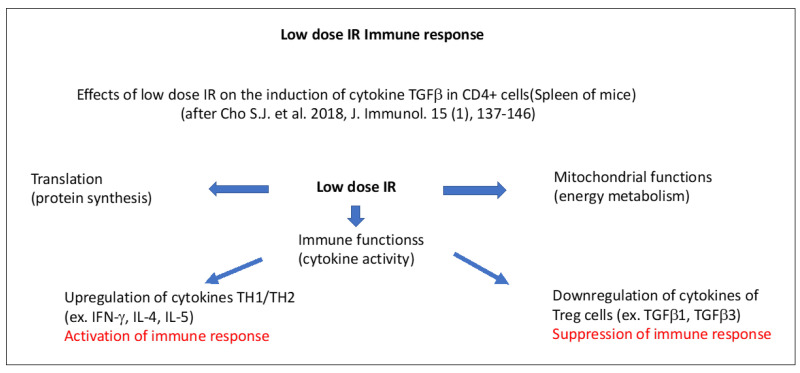
Schematic presentation of low-dose IR-induced immune effects in the spleen of mice, according to Cho et al. [274]. Distinct mitochondria-dependent pathways for the activation or suppression of immune effects are indicated. The upregulation of cytokines IFN-γ, IL-4, and IL-5 in TH1LTH2 cells leads to the activation of the immune response. The downregulation of cytokines TGFβ1, and TGFβ3) in T regular cells (Treg cells) leads to the suppression of the immune response.

**Figure 6 ijms-24-11460-f006:**
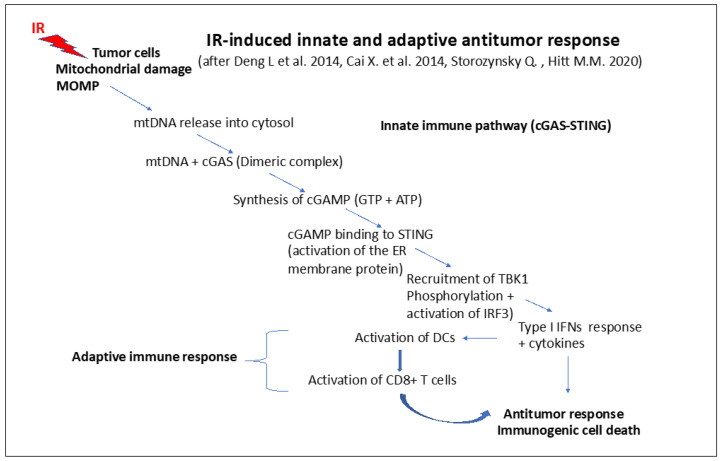
IR-induced innate and adaptive antitumor response (simplified scheme) inspired by [301,302,304]: steps of the innate and immune response pathway involving mitochondrial functions. Change of the mitochondrial outer membrane potential (MOMP) followed by the release of mtDNA into the cytosol and the formation of a dimeric complex between mtDNA and cGAS. This leads to the synthesis of cGAMP binding to STING, which recruits TBK1, activating the IRF3-dependent type I IFN response cytokine release, activating dendritic cells (DCs). DCs can, in turn, activate T cells (CD8+) that determine the immunogenic antitumor response.

## Data Availability

The data presented are taken from the scientific literature as referenced (Pubmed).

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
