# Peer review of "Low-Dose Non-Targeted Effects and Mitochondrial Control"

_ijms, 2023, doi:10.3390/ijms241411460_

Round 1
Reviewer 1 Report
This is an interesting review article by Dietrich Averbeck. The article involves a thorough discussion of the non-targeted and low-dose effects of radiation with an emphasis on mitochondrial functions. However, the article lacks some novelty crucial points in the subject of discussion here. I would suggest the following addition to this manuscript.
1- As most of the information is already available in the literature. The author needs to add some novelty in terms of perspective. For example, it has been discussed that low-dose radiation elicits both adaptive and harmful effects. Discussing what factors are possibly contributing to these opposite biological functions. This would be interesting and informative to the readers and researchers working in this area.
2- The author has given a detailed functional role of mitochondria and its components in the response to low-dose radiation. It would be good to discuss the therapeutic potential of mitochondria in radioresistance. How could the mitochondria and their components be leveraged in radiation therapy and in the mitigation of radiation injury?
3- The author needs to make a table with important aspects of the mitochondrial role in NTE following low-dose radiation, with relevant references in a column-by-column manner. This will be a snapshot of what has been detailed in the main text.
Author Response
Dear Reviewer 1,
Thank you very muc for your vary valuable comments. As you can see in the revised manuscript I added several new paragraphs in Chapter 10 (Concluding remarks).
In particular I added
- a chapter on low-dose radiation-induced adaptive (beneficial) and harmful effects in chapter 10 a new paragraph on ‘Factors possibly contributing adaptive beneficial and harmful effects of low doses’ lines 1345-1446-1488, ;;;;;with ‘biopositive effects’ lines 1455-1478 and ‘bionegative effect’s lines 1480-1488.
Factors possibly contributing adaptive beneficial and harmful effects of low doses
In recent years, it has become evident that low dose effects of IR(LDIR) can be adaptive beneficial as well as detrimental. Often, nonlinear effects have been observed at low dose exposures in contrast to high dose exposures for important biological endpoints [4]. In fact, they are often beneficial and more rarely harmful [66, 92, 340]. These possible dichotomic effects are less apparent at high doses where harmful effects follow a linear induction pattern and the LNT model is dominating.
The repair of DNA DSBs was not linear in the very low dose range. Moreover, DSBs induced by very low dose (1 mGy) of X-rays were efficiently repaired in proliferating normal human fibroblasts but not in quiescent fibroblasts [77].
Genetic factors involved in beneficial hormetic responses and harmful responses are partially overlapping [70]: LDIR-induced hormesis was shown to include ATM, extracellular signal-related kinase (ERK), mitogen-activated protein kinase (MAPK), phospho-c-Jun NH(2)-terminal kinase (JNK) and protein 53 (P53)-related signal transduction pathways. MAPK and p53 are also involved in adaptive responses. On the other hand, LDIR-induced bystander effects and genomic instability may include COX-2, ERK, MAPK, ROS, tumor necrosis factor receptor alpha (TNFα) and ATM, ERK, MAPK, P53, ROS, TNFα-related signal transduction pathways, respectively.
LDIR adaptive responses
Although LDIR effects are still controversial, without any doubt, LDIR can induce important adaptive responses in mammalian cells and in animals [341]. Park H.S. et al. revealed that chronic low-dose IR (LDIR) (10 and 50 mGy) and a challenging dose of 2 or 10 Gy resulted in increases of AKT, acinus protein via NF-kB activation in different human normal and tumor cells [342]. Clear differences in normal and tumor cell responses to LD and HD were observed , likely depending on AKT activation regulated by protein phosphatase 2 (PPA2). LD chronic exposure of normal cells lacking basal Akt activity increased activation of the ERK pathway involved in adaptive IR responses.
Regulators such as cyclin D1/CDK4 and cyclin B1/cyclin-dependent kinase 1 (CDK1) complexes are mediators between IR-induced DNA damage and mitochondrial functions regulated through phosphorylation of mitochondrial targets. They can lower genotoxic stress by adjusting mitochondrial metabolism and enhance cellular homeostasis. The cell cycle and cell-cycle-associated proteins are regulated by a dose of 200 mGy. High doses may give rise to p53 independent signaling and inhibit apoptosis and cell cycle progression via p21 phosphorylation, whereas LDIR may activate p21that can inhibit CDK1, CDK2 and CDK4/6 allowing cell cycle progression, IR-induced hormesis and adaptive responses. LDIR can also activate p53, p21 and apoptosis, thus inhibiting oncogenesis[341]. In this view, also hormesis can be achieved by enhancing DNA repair, ROS/RNS production, activating Nrf2 and NF-kB and increased antioxidant defenses. Moreover, cell proliferation may be enhanced through activation of signaling pathways (PI3K/AKT, Ras/Raf/ERK and Wnt/b/b-catenin, and the innate and adaptive immune system can be activated by stimulating cytokine production. Guéguen Y. et al. demonstrated that LDIR elicited DNA damage repair pathways (involving p53, ATM and PARP), the antioxidant pathway Nrf2 and the immune inflammatory response (NF-κB pathway), cell survival/death pathway (apoptosis), the endoplasmic response to stress (UPR response), and other cytoprotective processes including autophagy and cell cycle regulation [148].
Also, miRNAs play a role in LDIR responses. Wang X.C. et al. observed after 50 mGy adaptive apoptosis prior to 20 Gy in A549 lung cancer cells: 16 miRNAs were differently expressed and involved in this LDIR response [343].
After low dose exposures of human cells nuclear retention of cyclin D1 plays an important role in mitochondrial ROS mediated genomic instability [174]. Mechanistically, mitochondrial ROS perturb AKT-cyclin D1 cell cycle signaling through oxidation of PP2A leading to the accumulation of nuclear cyclin D1 and genomic instability [344].
LDIR of 20-500 mGy induced clusterin (a survival protein) which is involved in adaptive responses and radioresistance in cultured human cells and in mice in vivo. It is secreted after LDIR and is probably involved in the development of genomic instability, modification of intracellular communication by binding to cell surface receptors (TGF-b receptors) [345].
Ahmed K.M. et al. provided evidence for cooperative functions of ATM, ERK, and NF-κB in inducing a survival advantage in human keratinocytes through a radioadaptive response after LDIR treatment (100 mGy) with X-rays) [346].
Low dose HRS was observed in radiation-induced acute myeloid leukemia. HRS stimulated cell killing and Sfpi1 deletions, thus enhancing the cancer risk by altering the probability of Sfpi1 deletions to occur and persist [347]. HRS induction in rAML cells at low doses (60 mGy) involved oxidative stress and an increase in ROS in these hematopoietic cells [348].
As has been reviewed by Kabilan U. et al., LDIR can orchestrate hormesis including the factors p53, NRF2, signaling pathways ATM/ERK/NF-kB, PKC-p38MAPK-PLC and AKT/ERK/TNFa, FOXO3A and TGF-b, as shown at the transcriptional, translational and post-translational levels [150]. The physiological outcome depends on the balance between sustained damage and DDR signals. This persistent adaptive state may then trigger genomic stability and enhanced immune functions contributing to longevity and protection from cancer. These autors also reported that sensing and repair of DSBs was altered by lowering the translation factor elF4G1 targeting translation of BCRCA1 after IR. BCRA1 involved in error-free homologous recombination repair was shown to play a role in regulating transcripts of genes involved in DDR and DDR-signaling [150]. It was able to shift the balance of error-prone NHEJ towards error-free HR pathway of DSBs which can be regulated by 53BP1 and through elFG1 translation control. The authors suggested that translational reprogramming of DSB signaling and repair is part of LDIR responses that may lead to radiation hormesis associated with lower neoplastic transformation, suppression of tumorigenesis and extended lifespan.
As shown by Fernando C. et al., the effects of low dose radium alpha rays may be quite different in different species. Human keratinocytes showed radioresistance, whereas the embryonic Chinook salmon cell line (CHSE-214) was resistant to g-irradiation but exhibited radiosensitivity towards exposure to alpha particles [349].
Proteomic analysis of the bystander effects induced in chondrocytes by chondrosarcoma cells exposed to X-rays and C-ions at 100 mGy revealed about 20 proteins that are involved in oxidative stress responses (mitochondria), cellular motility and exosomes pathways [350]. The conditioned medium contained 40 modified proteins. In low dose (100 mGy) condition, DNA damage-responsive genes were increased in chondrocytes, and also a large cluster of proteins involved in stress granules with likely cell protective functions. Some bystander effectors showed specificity in terms of radiation quality, i.e. towards X-rays or carbon ions. While translational proteins were associated with both, antioxidant pathways and IL-12 were associated with X-rays, and G1/S and G2 DNA damage were specific to C-ion exposure [150].
Immune and anti-tumor LDIR effects
Concerning immunological effects, LDIR induces possible anti-inflammatory effects at low dose, expression of pro-inflammatory cytokines at moderate doses and immunosuppression after higher doses (precursor cell death, plus exacerbated innate immune responses [351]. ATM can trigger NF-kB activation together with nucleoplasmic shuttling involving the NF-kB essential modulator. The activation of NF-kB involves release of the complex with IkB kinase. Cytokine expression can activate NF-kB, and NF-kB can also be activated via TLRs (Toll receptors) by danger signals from dying cells. Activation of the immune response can be beneficial or harmful (detrimental).
At LDIR, possible anti-cancerogenic effects can be observed. In RT, anti-tumor responses may be supported by TLR agonists activation of NF-kB. Interestingly, LDIR at 50 mGy could activate NF-kB on two phosphorylation sites (Ser36 and Ser418) without inducing genomic instability probably due to efficient DNA repair [352].
Whole-body LDIR on metastatic mouse models induced anti-tumor responses via alterations of the immunosuppressive tumor environment leading to a reduction of pro-inflammatory Ly6chigh monocytes in APOE-/-mice [353, 354]. LDIR was upregulating selected immune components INF-g, IL-4 and Il-5, and cytokines released stimulated CD4+ cell T-cells [355]. Also, fractionated LDIR caused NF-kB upregulation.
Moreover, also tissue specificity plays an important role in LDIR effects. For example, low doses resulted in neurological effects in exposed individuals [356]. Indeed, cells are very sensitively reacting to low dose insults. However, the reactivity depends on the energetic and physiological state of the cell and also on the presence of concomitant insults from environmental factors.
Furthermore, in recent years stress granules have become a topic in cancer research [357]. They are involved in various tumor-associated signaling pathways, including cell proliferation, apoptosis, invasion and metastasis, chemotherapy resistance, radiotherapy resistance, and immune escape, and in bystander effects [358] and this, also at low doses [350]. Interestingly, in HNSCC cells, ceramide-enriched membrane domains contributed to targeted and nontargeted effects of radiation through modulation of PI3K/AKT signaling [359]. In radiosensitive SCC61 cells, NTE effects were brought about by the formation of an IR-induced ceramide-enriched domain. In radioresistant SQ20B cells such domain allowed phosphatidylinositol-3-kinase (PI3K)/ AKT signaling. Disruption of membrane lipid rafts led to the radiosensitization of these cells.
Thus, intracellular stress granules should also be considered in radiation research [360] because they are linked to the general stress-responsive intracellular and intercellular network including mitochondria [361].
Biopositive effects of LDIR
Moreover, mitochondrial functions can be activated by LDIR boosting mitochondria-dependent immune reactions and activation of Nrf2 [103]. Such immune cell activation concerns certain blood cells (lymphocytes and eosinophils) [104] as well as erythrocytes ([69].
Mobilization of cellular defenses (Antioxidants) and signaling systems recovery from cognitive and intellectual deficiencies were found in the case of Alzheimer [100-102].
Biopositive effects of LDIR on the immune system
NK cells can be activated by LDIR in mice [362]. LDIR by 75 mGy of X-rays upregulated the Th1 cytokines IL-1β, IL- 2, IFN- γ, and TNF-α and downregulated IL-10 pro-
duction on days 12, 16, and 20, whereas HDIR can inhibit the production of these cytokines. LDIR pretreatment protected the cytokine-producing ability of splenocytes on
days 12, 16, and 20 to some degree; however, this effect did not last up to day 24. LDIR induced NK cell activation also in vitro, most likely through the p38 mitogen-activated protein kinase pathway, the molecular mechanism of LDIR-induced antitumor immunity enhancement. Interestingly regulation of the Akt and 38 pathway could alleviate Alzheimer disease in Drosophila [98].
LDIR (0.1 or 0.2 Gy of X-rays) can inhibit metastases and trigger the cytolytic NK activity in Balb/c mice. However, the mechanism involved is not yet clear [363].
After repeated low-level, ten daily exposures of radiosensitive BALB/c or radioresistant C57BL/6 mice to 10, 20 and 100 mGy of X-rays, NK cell-enriched splenocytes obtained from the animals showed significant up-regulation of their anti-tumor cytotoxic function. Also, peritoneal macrophages exhibited cytotoxic effects on tumor cells and increased NO production [364].
Following human peripheral blood exposures to LDIR of 50 and 150 mGy , transcriptomic profiling of gene expression showed upregulation of many genes such as HLA-DQA1, HLA-DQA2, HLA-DQB2, HLA-DRB1, and HLA-DRB5 involved in antigen processing and presentation, immune system-related diseases and cytokine-mediated signaling [365]. This suggested boosting of the immune system. Positive immune-stimulatory responses were also found in isolated human primary monocytes with activation of toll-like receptors (TLRs), mitogen activated protein kinases (MAPKs) and NF-kB signaling, especially after LDIR exposure to low doses (0.05 and 0.1 Gy). P53 was not involved.
Beneficial effects of LDIR in combination with immunotherapy were also observed by. Barsoumian H. B. et al. who showed that LDIR enhanced systemic antitumor responses by overcoming the inhibitory stroma in established 344SQ lung adenocarcinoma in 129Sv/Ev mice [366].
Bionegative effects of LDIR
In mice, LDIR can be protective (91) but higher doses detrimental [91]. It can affect neurologic functions by downregulating neural pathways [99].
LDIR (g-rays) may program macrophage differentiation to an NOS (+)/M1 phenotype that orchestrates effective T cell immunotherapy [277], and the tumor environment is modified by retuning tumor-associated macrophages [366].
Recently, conditioned medium from irradiated WI-38 lung fibroblasts and H1299 lung adenocarcinoma cells exposed to 0.1 -1 Gy enhanced migration and invasion of unirradiated unirradiated H1299 cells without inducing apoptosis but senescence in a c-Myc-dependent way [368]. This suggests that the bystander responses can be dependent on the particular oncogenic state of the cells.
- the therapeutic potential of mitochondria in radioresistance: a new chapter on the ’ Role of mitochondria in radioresistance’ ,see lines 1586-1695.
Role of mitochondria in radioresistance
As already mentioned above, recognition of foreign particles and molecules in the cells is essential for cellular defense strategies [17,312]. For these, mitochondria have a central and prominent position. They are involved in cellular signaling of all types of damages, they are providing the energy for the different types of cellular responses, they direct innate and adaptive immune responses and coordinate long term responses and play a pivotal role by determining final adverse or beneficial outcomes.
Depending on the characteristic nature and type, dose and dose rate, IR is able to elicit in some particular circumstances damaging or beneficial outcomes involving specific mitochondrial reactivity. To some extent, also ATM is involved.
Albeit, the induction of radioresistance involving ATM, cyclooxygenase-2 (COX-2), ERK, JNK, reactive oxygen species (ROS) and P53 is somewhat ambiguous: in normal cells radioresistance is a biopositive effect, whereas in antitumor RT it is considered to be a bionegative effect and a serious drawback counteracting RT efficiency..
Since mitochondria constitute the main cellular powerstation it is not surprising that mitochondria are involved in radioresistance [395]. It is known for some time that radioresistance of cells involves mitochondrial glucose metabolism including glycolysis and oxidative phosphorylation [396]. And malignant transformation, tumor progression, and evasion of exogenous stress are influenced by mitochondria metabolism [397, 398].
Already Warburg found that cancer cells can undergo aerobic glycolysis with increased glucose uptakes, glycolysis and high lactic acid production [399].
Cancer cells are characterized by the mitochondrial synthesis of NADPH through the pentose phosphate pathway and the decrease of oxidative phosphorylation and the dependence of tumors on glycolysis [400]. In acidic condition of tumor environments cancer cells are able to reduce extracellular acidification and increase O2- production by switching from glycolysis to oxidative phosphorylation[396]. Thus, promoting tumor invasion and radioresistance to RT [401]. In fact, glycolysis is upregulated in most tumors without mitochondrial dysfunction. In these cancers, OXPHOS continues normally, even producing as much ATP as normal tissue at the same partial pressure of oxygen [402]. The kinase AKT can interfere with mitochondrial metabolism, enhance aerobic glycolysis and mediate radioresistance in human tumors [403].
Many inhibiting molecules have been developed against ROS and oncometabolites or to regulate OXPHOS and apoptosis, which can target specific receptors and enhance radiosensitization of tumor tissue. However, some are lacking specificity and have to be adjusted individually to the tumor type to overcome radioresistance [224].
Numerous reports show that metabolic inhibitors can interfere with mitochondrial metabolism and confer radiosensitization effects (see [316]. For example, inhibitors such as 2-deoxy-D-glucose of mitochondrial glucose metabolism radiosensitize cancer cells [404].
Oxidative phosphorylation inhibitors which affect mitochondrial function and reverse radioresistance [316] include metformin and phenformin affecting complex 1, and cervical cancer, head and neck squamous cell carcinoma, glioblastoma (IDH-wildtype), breast cancer,; arsenic trioxide (As2 O3 ), in the treatment of acute promyelocytic leukemia and recently radioresistant solid tumors, and some cancer cells of lung and liver. Also, Atovaquone an inhibitor of electron transport complex III significantly increased oxygenation and sensitized tumors to radiotherapy and radiosensitized hypopharyngeal, colorectal, and lung cancer cell lines. High-grade radio-resistant gliomas can be radiosensitized by dichloroacetate through activation of OXPHOS by reversing aerobic glycolysis [405]. Radioresistant cervical cancers were shown to be sensitive to inhibition of glycolysis and redox metabolism [406] .
RT itself affects mitochondrial energy metabolism, mitochondrial morphology and functions., and mitochondrial DNA mutation rates, respiration and ATP levels are increased. Changes in mitochondria membrane potential and in mitochondrial energy metabolism are primary events in tumorigenesis and radioresistance in RT.
Factors that affect MMP and confer radiosensitivity are for example, growth differentiation factor-15 (GDF15) belonging the TGF-b superfamily may represent a target to radiosensitize head and neck cancer cells by reducing MMP activation and allowing ROS generation, inhibitors of MEK/ERK-mediated signaling such as PD98059 increasing MMP and FAS-mediated cell death (and caspase-8 activity), inhibitors of histone deacetylase will reduce MMP and increase ROS generation together G2/M phase cell cycle arrest with apoptosis in esophageal cancers. Apparently, the blockage of the mitochondrial potassium -ATP (KATP) channel and ROS-induced MAPK/ERK kinase activation can radiosensitize glioblastomas [407]. For example, ROS generation and apoptosis could be enhanced in squamous cell carcinoma by poly-drug elevation of ceramide levels and radioresistance could be overcome [408].
Concerning ROS generation, supplementation of cancer patients with antioxidants can be detrimental with adequate antioxidant status (lung, gastrointestinal tract, head and neck and esophagus), but beneficial to individual cancer patients with deficient antioxidant systems [409].
Among the list of inhibitors of mitochondrial metabolic functions [316] are inhibitorsof glucose transporters, agents increasing ROS and oxidative stress, agents altering MMP, up-regulators of pro-apoptotic genes (BAC and BAX), inhibitors of NF-kB (for example curcumin). If ROS levels continue to increase beyond the antioxidant capacity of cells, this will cause apoptosis, ferroptosis, or cuproptosis, and significantly improve the efficacy of radiotherapy [398] .
Zaffaroni M. et al. revealed other promising inhibitors that can increase the effectiveness of RT against cancers (breast, brain, melanoma, prostate and ovary) [410]. One promising drug is lonidamine an inhibitor of aerobic glycolysis in cancer cells. It affects the succinate-ubiquinone reductase activity of mitochondrial complex II leading to enhanced ROS. Thus, glycolytic/mitochondrial metabolic changes appear to mediate cellular radioresistance.
A change in epigenetic regulation appears to be another important factor in radioresistance. Compared to nDNA, mtDNA is mostly hypomethylated. Epigenetic regulation is based on the mitochondria-specific DNA methyltransferase (mtDNMT). Under IR-induced oxidative stress CpG islets in mtDNA are oxidized and not available as methylation sites and mtDNMT is inhibited. This affects regulation gene expression and genome integrity. Moreover, it may interfere with the production of important co-factors, for example, ATP and acetyl-CoA involved in the acetylation of histones [6].
Due to the important role of mitochondria in metabolism and cell death [411] many factors are still to be discovered. One is papaverine an inhibitor of mitochondrial complex I which cause increased radiosensitization of solid tumors via oxygenation without important side effects in RT [412]. Another factor is pyrazinib which is radiosensitizing radioresistant oesophageal adenocarcinoma via the modulation of mitochondrial bioenergetics [413].
Undoubtedtly, mitochondria play a role in radioresistance, and adaptive responses of IR were observed in some tumors. For example, Aravindan N. et al. showed that LDIR can induce an adaptive response via the activation of NF-kB dependent responsive tumor necrosis factor a (TNF-a), interleukin 1a, cMYC and SOD2 via intercellular communication and sequential orchestration endorsing radiation protection (radioresistance) of surviving tumor cells [414].
Radioadaptive resistance of glioblastoma in RT involved the heme containing enzyme cytochrome C oxidase affecting the cellular iron pool and IR-induced Fenton reactions with hydroxyl radical production [415] . Interestingly, disruption of mitochondria was shown to radiosensitized prostate cancer cell lines[416]. Clearly, radioresistance of tumor cells can be reversed by targeting mitochondrial metabolism [316].
The direct implication of mitochondria in radioresistance has been also demonstrated by Grasso D. et al. [417]. They compared a radiosensitive SQD9 wildtype clone of HNSCC cells with a radioresistant SQD9 derived clone which harbored about 50% more mitochondria and a denser network around the cell nucleus and about 35 more mtDNA. Apparently, mitochondria protected against IR-induced damage. Thus, the targeting of mitochondrial metabolism remains a valuable option in anticancer RT [418].
Furthermore, also radiation quality determines how strongly RT can modulate mitochondrial functions and whether mitochondria can even mitigate radiation long term injury [19].
High LET IR, for example Carbon ion therapy (CIRT) is locally very damaging for mitochondria in particular in tumor cells but leave the tumor microenvironment and the surrounding immune cells largely intact to mobilize immune anti-tumor responses via dendritic and CD8+ T cells. Carbon ions are strongly inducing complete apoptosis and tumor cells death and also immunogenic tumor cell death. Instead, low LET IR appears to affect also the tumor environment together with the surrounding mitochondrial energy supply for surrounding immune cells. It follows that low LET IR is less immunogenic for tumors and needs some additional support through immunotherapeutic means (gene-mediated immune therapy, vaccine therapy, immune checkpoint inhibitors (anti PD1, anti-PD-L1) etc. [30].
- Concerning the question concerning important aspects of the mitochondrial role in NTE, I included a Table 1: Role of mitochondria in low dose IR nontargeted effects (NTE) in the final chapter 10. See “Role of mitochondria in radioresistance”
line 1299-1303.
Table 1: Role of mitochondria in low dose IR nontargeted effects (NTE)
|
IR Exposure Exp. Device |
Biological system |
Observation |
Reference |
|
Low LET photons (g-rays, X-rays,) (0.5, 5 and 7.5 Gy) ICCM |
Mammalian cells, Chinese hamster (CHO-K1) Human keratinocytes (HGV-G) Medium transfer |
Absence of RIBE in CHO-K1 mutants with nonfunctional glucose-6-phosphate dehydrogenase (G6PD) involved in mitochondrial metabolism- in HGV-G by inhibition of apoptosis and lactate metabolism. Alteration of calcium fluxes and loss of mitochondrial membrane potential (MMP). Involvement of mitochondrial ROS. |
[320] [321 ] [326]
|
|
Low LET IR 5 mGy and 0.5 Gy ICCM
|
Human keratinocytes (HGV-G)
Medium transfer |
Reduction of clonogenicity. Induction of increases in mitochondrial mass and low Bcl-2 expression in bystander cells after 5 mGy in ICCM but increased expression after 5 Gy. Decrease in survival. |
[314] |
|
X-rays 1Gy ICCM |
Human hybrids cells: Chinese hamster ovary (CHO) cells GM10115 + human chromosome 4, Medium transfer |
IR-induced mitochondrial dysfunction results in persistent high levels of ROS perpetuating genomic instability + clastogenic and transgenerational effects |
[250,251]
|
|
IR: g-rays 5 Gy ICCM |
Chinese hamster (CHO-K1) Human keratinocytes (HGV-G) Medium transfer |
Increase in mitochondrial mass, dysfunctional mitochondria in BE
|
[327] |
|
IR: g-rays (5 mGy, 0.5 Gy, 5 Gy) ICCM |
Chinese hamster (CHO-K1) Human keratinocytes (HGV-G) Medium transfer |
Mitochondria are sensitive to LDIR and ICCM, loss of enzymatic functions (OXPHOS), altered mtDNA-directed protein synthesis . |
[328] |
|
g- rays or 160 kV X-rays (0.5 Gy)
|
Human mammary epithelial cells (HMEC); Balb/c mice TGF-b1 +/- and +/+ |
Increased centrosome deregulation as a function of time. After clonal expansion CA increased in HMEC but unstable cells could be deleted by TGF-b1 via p53-dependent apoptosis (involving mitochondrial signaling) TGF-b1 can also suppress EMT. |
[329]
|
|
Microbeam IR with Carbon ions or X-rays
|
Mammalian cells Murine lymphoma L5178Y in co-culture with irradiated neoplastic epithelial cells.
Co-culture experiments
|
Cytoplasmic and cell irradiation affects mitochondria and calcium fluxes in targeted glioma and fibroblast cells Cytoplasmic IR involved mitochondrial damage and RIBE response. |
[245]
|
|
Microbeam with a-particles
|
T98G glioma cells and AG01522 fibroblasts
ICCM or Co-culture experiments |
Calcium signaling occurs early (RIBE). NO and mito-chondrial ROS lead to chromosomal damage (MN) |
[244]
|
|
241AM source a-particles (100 mGy)
ICCM |
Hamster normal AL cells r+ and mtDNA-depleted Medium transfer
|
Mitochondria-derived NO and O2- |
[330, 331] |
|
Microbeam IR with 1-10 protons ICCM |
Human keratinocytes HGV-G Medium transfer
|
ROS levels increased in bystander cells. Apoptosis induced was associated with a decrease in MMP and increased intracellular Ca2+ levels. |
[243] |
|
Microbeam IR with a-particles
|
Cervical cancer cells (HeLa) and mitochondria depleted pseudo-ρ0 cells
|
No RIBE in the absence of mtDNA. Signaling is inhibited by ROS and RNA inhibitors. Mt-dependent 53BP1 delocalization. BE involves intact mt signaling from targeted cytoplasm to the nucleus. |
[197]
|
|
Microbeam IR with 4He ions (120
|
Human fibroblast cells ρ0 and ρ+
|
High BE mutagenic response in mtDNA depleted ρ0 cells. BE involved mt-dependent NF-kB/iNOS/NO and NF-kB/COX-2/prostaglandin E2 signaling and NOS and COX2 signaling. |
[246] |
|
Low dose a-particles (0.29 mGy-25 mGy) and g-rays (2 mGy -50 mGy |
208F and v-src trans-formed 208Fsrc3 rat fibroblast cell lines. Co-culture experiments |
Low dose IR of non-transformed cells can induce apoptosis in precancerous cells through RIBE involving ROS/NOS signaling and cytokines such as TGF-b. The stimulatory effect saturates at 50 mGy for g-rays and at 25 mGy for a-particles |
[192] |
|
1GeV/u iron ions (LET ~ 151 keV/μ m), 600 MeV/u; silicon ions (LET ~ 51 keV/μ m), or 1 GeV protons (LET ~ 0.2 keV/μm). |
Normal human fibroblasts (AG1522)
Test of progeny: co-cultures of cells exposed to low or high doses of high LET IR |
RIBE depends on radiation quality and dose, and oxidative stress involving mitochondria |
[187] |
|
g-rays (0.05 and 0.5 Gy) ICCM |
Human keratinocyte cell line (HaCaT)
Medium transfer |
Low dose expression of genes involved in mitochondria-driven intrinsic apoptosis induced in bystander cells at low dose (50 mGy) |
[191] |
|
g-rays, a-particles and HZE particles (500 mGy) |
Normal human FB (AG1522 cells) co-cultured with a-irradiated HeLa cells (500 mGy) (connexin 32) Co-culturing |
Increased induction of MN and GI in bystander cells |
[225] |
|
Tritium (b-radiation) induced UV biophoton emission |
Human colon carcinoma cell line, HCT116 p53 +/+
Biophoton emission involvement in BE Exosomes |
Biophoton electromagnetic bystander signaling compromizes mitochondrial complex V (ATP production) and may be involved in the human fatigue syndrome. Exosomes extracted from UV-ICCM modulates clonogenic survival and MPP in bystander cells |
[38] |
|
g-rays (22 mGy) and biophoton emission |
Cells: HCT116 p53.+/+ Test involvement of cellular emissions of biophotons in gamma radiation-induced bystander cells |
Low dose biophoton emission from irradiated human cells may cause detrimental low dose RIBE. |
[39] |
|
6MeV photons (Clinac 600), 2 Gy |
Fadu cells derived from HNSCC
Secretion of exosomes in RIBE |
NTE is propagated by mtDNA and RNA in vesicles similar to exosomes |
[28] |
|
X-rays (0.1, 0.25, 2 Gy) Extracelluar vesicles (EVs) |
C57BL/6 mice
Total body IR, Extracellular vesicles (EVs) |
A panel of miRNA are involved in EVs bystander effects, differently at low and high dose, IR induced systemic effects |
[332] |
|
X-rays 4 Gy CCCM/ICCM |
Seven-week-old male ICR mice:
ELV from irradiated mouse serum and ICCM |
Absence of DNA damage in CCCM ELV or ICCM ELV from mt-depleted ρ0 normal human fibroblasts. Secretion of mtDNA via exosomes is involved in mediating RIBE signals. |
[32] |
|
X-Rays Partial and whole body exposure 2 Gy |
C57Bl/6 female mice of eight weeks of age Analysis of ‘Out of field ‘effects partial body IR in mice
Exosomes |
Deregulation of many proteins and miRNAs. Some miRNA, proteomic changes and exosomes are involved in anti-apoptotic effects. Injection of exosomes from irradiated mice can prevent apoptosis. |
[227-228]
|
|
g-rays, high doses (2-8 Gy) ICCM |
Human HepT2 cells
Medium transfer from irradiated cells |
Induction of Bax, Bcl2, caspases and g-H2AX DNA damage in bystander HepT2 cells |
[333] |
|
200 kV X–rays (6 Gy) |
Human pancreatic cancer cells (MiaPaCa–2), wild–type (wt) and ATM−/− fibroblasts
Co-culturing |
Healthy ATM +/+ cells modify the DDR of irradiated cells by a microtubule- and ATM-dependent exchange of healthy mitochondria.
|
[202] |
Hoping that you also find these additions interesting and useful, With many thanks to you and with
my best regards,
Dr. Dietrich Averbeck
Reviewer 2 Report
This is an impressive and comprehensive review that clearly provides a new perspective on Non-Target Effects of radiation focused on the role of mitochondria. I found the argument persuasive, the insights very useful and the manuscript a pleasure to read. To me, it is high quality manuscript that should be published.
I have only some general comments that might be of some use. I don't think they need to be incorporated into the manuscript but if they author finds them helpful then please use them, if not they can be ignored.
1. Dose rate dependency. I realise that dose rate is mentioned several times but it doesn't feel to me that dose rate effects are really scrutinised in the paper. The key insights hinge, primarily I think, on total doses. I think that what is known of the dose-rate dependency of the effects would be useful to highlight. I'm not a biochemist and I also realise that much of the literature is studies related to RT in which the focus is usually on total doses, but I've a feeling the role of mitochondria in the responses might be a good example of a phenomenon in which dose rate is important. There are of course molecular switches and other somewhat binary phenomena involved but there is much rate-dependent biochemistry and I wonder if a bit more focus on dose rate at points might help the argument for the role of mitochondria in the responses. Perhaps it's just a personal delusion but I've feeling that dose rate is often underestimated on discussions of phenomena such as those discussed in the review and might be worth a bit more of an explicit mention at points. Biochemistry is dynamic and rate dependent so 'stress rate' matters I feel.
2. The biology of mitochondria. At places a bit more mention of wider mitochondrial biology might be helpful. Mitochondria are fairly similar throughout the eukaryotes but not identical and different cells types vary a great deal in how many mitochondria they have. If mitochondria have the roles suggested in the review, any comparisons with organisms that have slightly different mitochondria or that have no immune systems might be insightful? Mitochondria also have a biology that reaches back into prokaryote evolution which first happened at a time with higher background IR and probably a great deal of UV, some of which was ionising. Does this provide any insights? Do differences in DNA in mtDNA as compared to the nuclear DNA have any relevance? Are mitochondria more radioresistant than other organelles because of their history?
The MS is very well written and there are very few typographical errors but there a some to be amended.
Author Response
Dear Reviewer 2!
Thank you very much for your very valuable comments. I tried my best to revised the manuscript accordingly.
Concerning the biology of mitochondria: I tried also in the chapter 10. “concluding remarks” to focus on the importance of the mitochondria. For this, I included a new paragraph lines 1225-1249, and another paragraph 1255- 1291 on the importance of bystander and NTE involving mitochondrial functions (RIBE and NTE and the role of mitochondria).
1228-1249
The largest part is involved in mitochondrial energy supply and cellular metabolism. Electron leakage during energy metabolism leads to the generation ROS and NOS [15]. Mitochondrial ROS are usually 10 times higher in tumor than in normal cells [311]. To avoid excessive oxidative damage cancer cells activate potent cellular antioxidant systems to counteract ROS by superoxide dismutases (SODs) enzymes in mitochondria, catalyzing O2- to H2O2. These can be reduced to H2O by catalases (CATs), glutathione peroxidases (GPXs), and peroxiredoxins (Prxs) [311,312].
The highly diffusible reactive oxygen or nitrogen species (ROS and NOS) are formed during metabolic activity and are involved in redox regulated signaling. These are second messengers in cell signaling and direct gene expression. Indeed, oxidation reactions promote activation of protein kinases, whereas phosphatases and zinc finger proteins are inactivated. Transcription factor activation of NRF2 ((nuclear factor erythroid 2–related factor 2) is increased by reduction reactions. Redox sensitive reactions of signaling proteins are often reversible in order to allow switching and adjustment of metabolic activities. The levels of antioxidants determine the outcomes of IR [313].
Moreover, important structural differences exist between mitochondrial DNA (mtDNA) located in the inner membrane space of mitochondria and nuclear DNA. Mitochondrial DNA is more radiosensitive because of the lack of protective histones and limited DNA repair [293]. Furthermore, mtDNA can be released by dying cells and is part of damage-associated molecular patterns (DAMPS ) [298] which are very immunogenic because of the presence of CpG isles.
Since mitochondria constitute the energy platform [253,268,287,291] the first reaction after IR exposure is a modification of the energy metabolism. Generally, metabolically very active cells such as cancer cells contain more mitochondria and more mtDNA than normal cells. Mitochondrial mass varies in different species and animal organs, mostly related to their proliferative capacity. IR exposures at low (5 mGy) and high doses (5 Gy) can produce increases in mitochondrial mass [314].
1243-1249
Low doses induce already small but significant changes in oxidative phosphorylation resulting in leakage of ROS and NOS and changes in metabolic ATP production. This affects the energy available to the targeted cells and other bystander cells. A special case are tumor cells in interaction with the tumor micro-environment.
As reviewed by Klammer et al., there are many factors involved in radiation-induced bystander (RIBE) and NTE effects: the involvement of mitochondria related functions and bystander and innate and adaptive responses is striking[193]. ROS and mitochondrial generated ROS are very important and determine the oxidative state and oxidative stress level induced. Modulation of oxidative stress by IR is crucial for the bystander and also for the immunological responses. Some signaling cascades (Calcium fluxes, MAPK and NF-kB networks) participate in both phenomena and appear to be associated. Bystander effects can perpetuate oxidative stress induced damage and also immunological (inflammatory) responses which may continue to perpetuate damage. Oxidative stress and oxidative damage are highly damaging and drive immunological responses through DAMPS and PAMPS [224,315]. According to Kong et al. mitochondrial ROS and ATP can be considered as damage-associated molecular patterns (DAMPs) that may give rise to systemic inflammatory responses [224].
In normal cells, ROS are produced at low concentrations and are effectively neutralized by the potent antioxidant systems of the cells [316]. A moderate increase in ROS levels by chronic oxidative stress and LDIR induces random mutations in cells and promotes tumor cell proliferation, metastasis, and radioresistance. Moderate ROS may increase radioresistance of cells in RT by triggering adaptive hometic responses and promoting autophagy [317] or trigger apoptosis independently of DNA damage [316,318]. ROS basal levels are often higher in tumor than in normal cells [319]. Low to moderate ROS levels act as signal transducers activating cell proliferation, migration, invasion and angiogenesis, whereas high ROS levels damage proteins, nucleic acids, lipids, membranes and organelles associated with cell death [319]. Manipulations of IR-induced mitochondrial ROS are promising in RT and immunotherapies.
1255-1291
RIBE and NTE and the role of mitochondria
A number of valuable reviews describe and comment on the importance of ionizing radiation- induced bystander (RIBE) and nontargeted effects (NTE) in ionizing radiation responses concerning initially the effects of low dose ionizing radiation (LDIR) (< 100 mGy) and the effects of moderate and high dose (RT) dose exposures in recent years. At first, the bionegative effects of LDIR induced bystander effects such as cytotoxicity [78, 314, 320, 321], mutagenicity [ 244, 187, 248], genomic instability [2,3, 174, 225, 233, 250, 251, 259, 322], cancerous effects [226] and inflammatory effects [224] were in the center of discussions, also because of nonlinear responses that were incompatible with the LNT model. Later, closer insights revealed that bystander effects and NTE could also exhibit biopositive effects [29, 70, 72, 107, 130, 192] covering all dose ranges with systemic and longer distant abscopal reactions mediating biopositive effects by modulating innated and adaptive immune responses with favorable outcomes in anti-pain and anti-cancer RT [227-228, 230, 232, 235, 278] but sometimes also bionegative [229]. As shown in this paper, a sort of red line is constituted by mitochondria involved in most IR-responses.
The following underlines that bystander and NTE mainly rely on mitochondria driven energy metabolism, mitochondria-dependent apoptotic signaling, mitochondrial reactivity to calcium fluxes, changes in mitochondrial membrane potential, mitochondrial ROS and NOS generation, mitochondrial DNA, mitochondria-dependent 53BP1 delocalization, cytokine and TGF-b release, mitochondria -dependent NF-kB/ iNOS/NO and NF-kB/COX-2/prostaglandin E2 signaling pathways, oxidative status of the bystander cells, level of oxidative stress induced by IR of different radiation quality (LET), IR dose level, radiation-induced biophoton level (biophoton emission of irradiated cells), IR-induced exosomes and their cargo (mtDNA, types RNAs, nDNA, cytokines etc.).
The questioning on the different effects of UV and IR on mitochondria is in my point of view partially answered by the phrase “UV-induced bulky adducts in mtDNA such as prymidine dimers are not repaired in mitochondria since they are lacking the nucleotide excision-repair system[293] (see lines 1021-1022). The differences in reactivity of nDNA and mtDNA are allotted to in lines 1018-1021, line 1244, 143-1046, line 1665 showing that mtDNA is easily repairable.
- The question “Are mitochondria more radioresistant than other organelles because of their history ?” I cannot answer at present. Part of the answer may be based on the mitochondrial dynamics (referred to in lines 665-668 ) and on the high plasticity of mitochondria (referred to in lines 1330-1332).
- The dose rate dependency is now described in a large part of chapter 10 (lines 1490-1550) with emphasis on the modulation of mitochondrial biogenesis, anti-inflammatory responses and nonlinearity in the low dose rate range as well as the effects of chronic LDR exposures.
Low dose-rate effects
Low dose-rate (LDR) effects play an important role in low dose effects of IR. As recalled by Amundson S.A. et al., classical observations on protracted low dose radiation exposures yield in normal, DNA repair competent cells important sparing effects because of efficient DNA repair [369]. LDIR go often together with LDR exposures (see radioadaptive responses). LDR exposures are generally protective against mutation induction and cell transformation in vivo and in vitro. As shown by Rothkamm and Löbrich, at very low dose 1 mGy of IR on proliferating cells high dose-rates rapidly turn on the DDR pathway involving ATM and phosphorylation of H2AX (gH2AX) [77]. In resting cells this was not the case, and DNA repair was stalled at (1-20 mGy). Collis S.B. et al. were able to show in cancer cell lines that at low dose-rate IR (i.e. 450times lower than a high dose rate producing environ 4-5 DSB/ h instead of 1800 DSB/h) increased cell killing (clonogenicity) as a consequence of inefficient activation of the DNA damage sensor ATM and H2AX phosphorylation [370]. Thus, DDR signaling is nonlinear at LDR. Even though, also inverse dose-rate effects have been observed with increased mutagenicity in somatic and germ cells due to cell cycle dependent radiation sensitivity windows. DNA repair deficient cells, for example. Fibroblasts from AT- patients deficient in ATM exhibit little or no dose-rate effects. Gene expression studies revealed that low dose-rate exposures triggered protection against protection against the induction of apoptosis with a linear induction o p53 regulated genes, except MDM2 and cell cycle regulated genes.
Very importantly, dose-rate is an important factor in modulating mitochondrial biogenesis [14]. A low dose of 0.1 Gy at a dose-rate of 0.055 Gy/min caused dysfunction of the mitochondrial respiratory chain in the rat small intestine enterocytes with perturbance of cytochromes in the inner mitochondrial membrane and inhibition of H+- ATPase activity [371].
Barjaktarovic Z. et al. showed that whole body IR of ApoE-/- mice (deficient in cardiac mitochondrial protein (associated with metabolic impairment and sirtuin downregulation) at chronic exposure at 20 mGy/day for 300 days induced increased acetylation and reduced mitochondrial sirtuin [372]. The hyperacetylation involved the mitochondrial TCA cycle, fatty acids oxidation, oxidative stress and the sirtuin pathway. Acetyl-CoA increased and cardiac metabolic regulators (PGC-1 alpha and PPAR) were inactivated.
- Interestingly, LDR exposures of normal human cells (48BR) induced activation of mitochondria-dependent oxidative stress involving AMPK, p38, MAPK and ERK but this was not seen in cancer cells [26].
LDR exposures are important in adaptive responses: for example, Sugihara T. et al. found that a low priming dose at low dose-rate (20 mGy/day) allowed an adaptive response 12 days later to a challenging dose of 6.75 Gy at a high dose-rate [373].
Several excellent recent reviews [374-376] give a detailed account of the key events of LDR effects in animal models. Prolonged life-times of mice after chronic low dose-rate IR exposure could be observed [377, 378]. However, Ogura K. et al. reported an increased copy number variation (deletions) in the offspring of male mice exposed to LDRIR associated with a possibly shorter life span [379]. Also, 3Braga-Tanake I 3rd et al. reported that chronic 1 mGy/day exposure of mice yielded significant changes in lifespan, neoplasm incidence, chromosome abnormalities and gene expression [380]. In fish cells, a low dose rate of 83 mGy/min exposure involved a biphasic response leading to a higher clonogenicity than after high dose-rate 366 mGy/min [381]. Nonlinear responses to low dose-rate exposures were observed in epithelial cells of the lens [382] and in human umbilical endothelial cells [383,384].
Also, immunological IR responses are affected by LDRIR. For example, Ina Y, K. Sakai showed that exposure of wildtype mice to chronic radiation 1.2mGy/hour increased CD4+T cells and CD8 molecule expression while CD40+ B cells decreased [385]. Chronic exposure at LDR activated the immune system of the whole body. They also reported that IR-induced lymphoma at high dose-rate was suppressed by pretreatment with low dose radiation at 75 mGy (adaptive response) and further repressed by lifelong g-IR at LDR 1.2 mGy/h (which on its own did not yield lymphomas or other tumors) [385].
Rey N. et al. showed a decrease of proinflammatory Ly6CH monocytes [386], and Edin NJ. et al. (2015) an activation of TGF-b3 at LDRIR [387].
LDR exposure (5 mGy/min) for 1 h induced protection against lethal IR doses effects without affecting life span of DBA/2 mice[388]. In human fibroblasts genes against oxidative stress were upregulated at low dose rate [389] .
Recent findings reveal that the main target of chronic ionizing radiation is the activation of the inflammatory system which can lead to the initiation of related processes such as apoptosis, cell differentiation and proliferation, angiogenesis, invasion and metastasis in tumor progression [390].
In addition, dose-rate and dose fractionation and FLASH exposures [391, 392, 393] determine the biological consequences and outcomes of IR. Interestingly, FLASH with protons (FLASH-RT) prevented mitochondria damage characterized by morphological changes, functional changes (membrane potential, mtDNA copy number and oxidative enzyme levels) and oxyradical production [393]. The Dynamin-1 like (Drp1) protein mediated mitochondrial homeostasis in FLASH-RT [391, 392, 393].
This also confirms that not only radiation dose but also radiation dose -rate and radiation quality are important for IR-induced biological responses and RT. In fact, low dose and low dose-rate IR effects have open many new of avenues in radiation biology that turn out to be very beneficial for recognizing and understanding important networks involved in LDIR and LDRIR responses that pave the way for better radiation protection and anti-cancer radiation therapies.
Hoping that you also find these additions interesting and useful,
With many thanks and
with my best regards,
Dr. Dietrich Averbeck